

# Intercomparison of Four Tropical Cyclones Detection Algorithms on ERA5

Stella Bourdin[1], Sébastien Fromang[1], William Dulac[2], Julien Cattiaux[2], and Fabrice Chauvin[2]

[1]Laboratoire des Sciences du Climat et de l'Environnement, LSCE/IPSL, CEA-CNRS-UVSQ-Université Paris-Saclay, Gif-sur-Yvette, France
[2]Centre National de Recherches Météorologiques, Université de Toulouse, Météo France, CNRS, Toulouse, France
**Correspondence:** Stella Bourdin (stella.bourdin@lsce.ipsl.fr)

**Abstract.** The assessment of Tropical Cyclones (TC) statistics requires the direct, objective and automatic detection and tracking of TCs in reanalyses and model simulations. Research groups have independently developed numerous algorithms during recent decades in order to answer that need. Today, there is a large number of algorithms, often referred to as trackers, that aim to detect the positions of tropical cyclones in gridded datasets.

This paper compares four trackers with very different formulations in detail. We assess their performances by tracking tropical cyclones in the ERA5 reanalysis and by comparing the outcome to the IBTrACS observations database.

    The first section of the paper finds typical detection rates of the trackers ranging from 75 to 85%. At the same time, false alarm rates (FAR) greatly vary across the four trackers and can sometimes exceed the number of detected genuine cyclones. Based on the finding that many of these false alarms are extra-tropical cyclones, we adapt two existing filtering methods

common to all trackers. Both post-treatments dramatically impact FARs, which range from 9 to 36% in our final catalogs of tropical cyclones tracks. We then show that different traditional metrics can be very sensitive to the particular choice of the tracker, which is particularly true for the TC frequencies and their durations. By contrast, all trackers identify a robust negative bias in ERA5 tropical cyclones intensities, a result already noted in previous studies.

    We conclude by advising against using as many trackers as possible and averaging the results. A more efficient approach

would involve selecting one or a few trackers with well-known properties.

## 1 Introduction

Assessing whether and how Tropical Cyclones (TC) activity will evolve with climate change is a crucial but difficult question to tackle. Because the theoretical understanding of these events remains incomplete, and the observations' timespan is too short to infer robust trends in their properties, projections of TC activity typically rely on model simulations (Knutson et al.,

2019, 2020). In this realm, the main bottleneck impediment is their limited spatial resolution, which is currently around $100\,\mathrm{km}$ for the vast majority of CMIP6 models. This resolution is still too low to simulate realistic tropical cyclones (Camargo and Wing, 2016; Roberts et al., 2020a). However, with the recent advances in computational resources, global simulations with atmospheric spatial resolutions that reach 50 to 25 kilometers are now feasible and will become more and more common in the future. The few high-resolution model results already published clearly demonstrate a dramatic improvement in simulating





TCs (Manganello et al., 2014; Murakami et al., 2015; Walsh et al., 2015; Roberts et al., 2020a). This avenue is raising hopes
in our capacity to better understand these storms and to better predict their future evolution.

Studying TCs in global simulations spanning several decades requires their objective and automatic detection and tracking,
which is accomplished by so-called TC trackers. Trackers are algorithms that are able to detect cyclonic structures associated
with a warm core in a gridded dataset and link them together into a trajectory. Many modeling and operational centers have

developed such trackers independently, and there is now a wealth of such algorithms available to the community and described
in the literature (see for example the list compiled by Zarzycki and Ullrich, 2017, in the appendix of their paper). Broadly
speaking, TC trackers can be divided in two main categories: "physics–based" and "dynamics–based" trackers. The former
are based on the detection of a local sea-level pressure minimum combined with a warm-core criterion (usually expressed
as a temperature anomaly or a geopotential thickness), on top of which discriminating intensity criteria are applied based on

surface winds or vorticity. This category includes for example the trackers from Camargo and Zebiak (2002), Zhao et al. (2009),
Murakami (2014), Horn et al. (2014), referred to as CSIRO, or Chauvin et al. (2006) and Zarzycki and Ullrich (2017), hereafter
referred to as CNRM and UZ respectively. "Dynamics-based" trackers, on the other hand, rely on dynamical variables such as
vorticity or other derivatives of the velocity. They include the TRACK method (Strachan et al., 2013; Hodges et al., 2017) and
the OWZ algorithm (Tory et al., 2013b). Trackers in the latter category often claim to be resolution-independent (Tory et al.,

2013a). By contrast, the physics-based trackers usually embed a wind threshold, a parameter known to be very sensitive to
resolution (Walsh et al., 2007).

Despite this diversity, only a few studies explicitly aim to compare different TC trackers. Horn et al. (2014) were the first
to put forward the question of trackers comparison. The authors showed that the results obtained using four physics-based
trackers could vary significantly because of the different thresholds and criterion variables used by the different algorithms.

Raavi and Walsh (2020) later performed a similar comparison between the CSIRO and the OWZ trackers. OWZ was found
to produce better results across a wide range of resolutions, while the CSIRO tracker performed better for the high-resolution
datasets.

These studies confirm the naive expectation that different tracking algorithms inevitably have different TC detection skills.
As a result, it is often difficult to compare different studies because they use different trackers. For example, future projections

of TCs frequencies in CMIP5 as reported by Tory et al. (2013b) and Camargo (2013) are difficult to compare because they
used the OWZ tracker and that of Camargo and Zebiak (2002), respectively. Two recent papers by Roberts et al. have tried to
circumvent this problem using multiple trackers when analyzing a given dataset and check whether the result is robust, i.e.,
independent of the tracker (Roberts et al., 2020a, b). These intercomparisons of a series of HighResMIP simulations (Haarsma
et al., 2016) use TRACK and UZ. In both papers, the authors reported large differences between the two trackers in the

frequencies of TCs. Nevertheless, they also confirmed robust improvements in TC statistics with spatial resolution regardless
of the tracking algorithm they considered. However, a detailed comparison of the two trackers' properties is still lacking at these
high spatial resolutions and would improve interpretations of modeling results. The present work performs such a comparison
in order to document the relative strengths and weaknesses of the large variety of trackers presented above, as well as provide
guidelines for the use of TC trackers in climate simulation outputs.



This paper reports the results of an intercomparison of four different trackers with properties as different as possible from one another in terms of their formulation. The report is based on a comparison between the tracks detected by these trackers on a reanalysis (ERA5, Hersbach et al., 2020) and an observation database (IBTrACS, Knapp et al., 2010). This study uses the reanalysis as a bridge between observations and simulation. Our main goal is not to provide an assessment of ERA5 performances in reproducing a given TC climatology but to compare the trackers with one another. Numerous studies have

undergone such an assessment on several other reanalyses, including ERA5's predecessor ERA-Interim (Hodges et al., 2017; Schenkel and Hart, 2012; Murakami, 2014; Bell et al., 2018). Only recently, Zarzycki et al. (2021) presented an evaluation of ERA5's TCs against other reanalyses. The study shows that ERA5 performs as well as reanalyses that include specific TC assimilation techniques such as JRA and NCEP and that a significant improvement is brought by the increase in resolution between ERA-Interim and ERA5. A comprehensive assessment of TCs in ERA5 will be presented in a forthcoming paper by

Dulac et al.

    The paper is organized as follows. After a description of the classification and datasets we used, we detail the algorithm of the trackers we used as well as our track matching method (section 2). We then use the four trackers to track TCs in ERA5 and to match the detected tracks with IBTrACS tracks, and we present a detailed analysis of the population of missing and false alarm tracks so obtained (section 3.1). This knowledge is used to develop two methods common to all trackers that aim

at filtering extra-tropical false alarms from the results (section 3.2 and 3.3). The filtered datasets are then used to analyze the sensitivity of traditional metrics to the choice of the trackers (section 4). Finally, we use the insight gained from this analysis to consider the complementarity of different trackers and provide some guidelines for applying TC trackers to model results (section 5). The conclusion gives a summary of the trackers' common points and differences (section 6).

## 2   Data and methods

Our analysis combines resources available on both the database of observed TCs IBTrACS (Knapp et al., 2010) and the ERA5 reanalysis (Hersbach et al., 2020). Before describing these two datasets in detail, we first highlight the procedure we used to classify TCs according to their intensities. We next describe the specifics of the four trackers we compare in this paper, and explain our track matching method.

### 2.1   Tropical Cyclones Intensities and Classification

TCs are commonly classified on the Saffir-Simpson Hurricane Scale (SSHS) with the peak 1–minute near-surface wind (generally at 10 meters above the surface). This is different from the World Meteorological Organization (WMO) standard to report the 10–minutes near-surface sustained wind $u_{10}$. For that reason, we have chosen to systematically convert 1–minute sustained winds to 10–minutes sustained winds. To do so, we used the 1.12 coefficient provided by the IBTrACS documentation (Knapp et al., 2010), although we note there are some ambiguities in the precise value one should use for that purpose (Harper et al.,

2010). As a result, $u_{10}$ must exceed $29\,\mathrm{m\,s^{-1}}$ for a given structure to be classified as a TC, while Tropical Storms (TS) are



| Category | Saffir-Simpson maximum 10-minutes wind threshold / $\mathrm{m\,s^{-1}}$ | Klotzbach minimum sea-level threshold / $\mathrm{hPa}$ |
|---|---|---|
| 0 (TS) | 16 | 1005* |
| 1 | 29 | 990 |
| 2 | 38 | 975 |
| 3 | 44 | 960 |
| 4 | 52 | 945 |
| 5 | 63 | 925 |

**Table 1.** Tropical Cyclones intensity classification. Saffir-Simpson Hurricane Scale thresholds are converted into 10-minutes sustained wind using a 1.12 conversion coefficient. *This threshold is not in the original classification but has been derived by ourselves using the same method.

defined as storms for which $16\,\mathrm{m\,s^{-1}} < u_{10} < 29\,\mathrm{m\,s^{-1}}$. The threshold values of $u_{10}$ for each TC categories are reported in table 1.

In the present paper, we will evaluate TCs intensities using their Minimum Sea Level Pressure (MSLP). As discussed in the literature in the past few years, the rationale behind this practice is twofold. First, MSLP is easier to measure than $u_{10}$
(Klotzbach et al., 2020), thereby reducing the uncertainty associated with its evaluation. It is also uniformly defined among the different forecast agencies (Knapp et al., 2010), thereby removing the uncertainties associated with the conversion between winds obtained for different averaging periods such as described above. In addition, models tend to be able to reproduce the observed range of TCs MSLP but fail to simulate the largest wind speeds (Knutson et al., 2015; Chavas et al., 2017). MSLP is a more reliable indicator of TC intensities than wind speeds. This is true in models, but also for ERA5, as recently shown by
Zarzycki et al. (2021). Finally, and even if we do not tackle TC damage in this study, MSLP has also been argued to be a better predictor of TC damage than maximum wind speed (Klotzbach et al., 2020).

Simpson and Saffir (1974) provided a version of the SSHS categorization in terms of pressure, but it does not preserve the proportion in categories of the wind scale. Therefore, we instead use the classification from Klotzbach et al. (2020) to compute TC intensity categories. It is reported in Table 1 for completeness.

## 2.2 Datasets

### 2.2.1 IBTrACS

The International Best Track Archive for Climate Stewardship (IBTrACS, Knapp et al. 2010) version 4 is the most comprehensive database of observed TCs. We used the "since 1980" subset in the present paper (Knapp et al., 2018). It combines data provided by TC centers of the World Meteorological Organization (WMO), namely the Regional Specialized Meteorological
Centers (RSMC) and Tropical Cyclone Warning Centers (TCWC), as well as non-WMO centers, such as the China Meteorological Administration, the Hong Kong Observatory, and the Joint Typhoon Warning Center. Because IBTrACS sources are so





diverse, the database is heterogeneous and requires careful treatment before one can safely use it. The steps we followed are summarized below and detailed in a workflow chart (fig. B1).

This study considers the cyclonic seasons from 1980 to 2019 in the northern hemisphere (NH, 40 seasons) and from 1981
to 2019 in the southern hemisphere (SH, 39 seasons). We removed seasons after 2019 because they contain provisional tracks. We also filtered out all tracks labeled as "spur" since they correspond to "usually short-lived tracks associated with main track and often represent alternate positions at the beginning of a system [or] actual system interactions"[1]. In the remaining tracks, we only kept 6-hourly time steps for consistency with ERA5. Winds and sea-level pressure (SLP) data were retrieved when available, prioritizing the WMO center responsible for the relevant region. Tracks lacking wind data (0.5% of all tracks) were
dropped. Tracks lacking SLP data (7% of tropical-storm intensity tracks) were kept but not be included in those parts of the analysis for which storm intensities are needed. Finally, we removed tracks that do not reach the tropical storm stage ($16\,\mathrm{m\,s}^{-1}$) and those that last less than one day.

Hereafter, our selection of IBTrACS data will be referred to as IB–TS. We also define IB–TC as the subset of IB–TS tracks that reached the tropical cyclone intensity ($u_{10} > 29\,\mathrm{m\,s}^{-1}$). IB–TS (resp. IB–TC) contains 3519 (resp. 1938) tracks.

### 2.2.2   ERA5

We used data from the fifth generation of ECMWF Reanalysis (ERA5, Hersbach et al. 2020). ERA5 provides hourly estimates of atmospheric variables on a grid with $0.25°$ horizontal resolution from 1979 to the present day. For the purpose of this work, we only used 6-hourly data from 1980 to 2019 (as in IBTrACS). Unlike other reanalyses such as JRA-55 or NCEP-CFSR, ERA5 does not perform any specific assimilation for Tropical Cyclones (Hodges et al., 2017). Nevertheless, it has recently
been assessed as having similar performances for a range of metrics (Zarzycki et al., 2021; Roberts et al., 2020a). These results motivated our choice to use ERA5 as a testbed to benchmark the detection skills of the four different TC trackers we will now describe.

### 2.3   TC trackers

We provide in table B1 a synthesis table of the trackers' criteria and thresholds presented below.

### 2.3.1   TempestExtremes

TempestExtremes (see https://climate.ucdavis.edu/tempestextremes.php) has been developed by Ullrich and Zarzycki (2017) as a command-line software enabling a fast and versatile implementation of TC trackers. For the tracking of pointwise features, such as TCs, it provides two functions: DetectNodes that find candidates "nodes" corresponding to local extrema of a given variable, and optionally satisfying a set of additional criteria (closed-contours, thresholds); and StitchNodes that links
candidates within a given distance of one another into a track. In this paper, we use TempestExtremes to implement two vastly

---

[1]IBTrACS columns documentation





different TC trackers, UZ and OWZ, respectively described by Ullrich et al. (2021) and Tory et al. (2013d). We describe both algorithms below and provide the associated codes in the appendix C.

### 2.3.2 UZ algorithm

We implemented the physics-based algorithm UZ in TempestExtremes as described by Ullrich et al. (2021). The thresholds
were calibrated by Zarzycki and Ullrich (2017) using sensitivity analysis to several metrics and the data of four reanalysis products. This tracker was referred to as "TempestExtremes" in Roberts et al. (2020a, b) but we prefer to distinguish between the framework and the tracker formulation itself.

**Candidate detection:** The first step consists in finding the local minima of the sea-level pressure (SLP). It defines a series of candidate points. In a second step, only those candidates that verify the following two closed-contour criteria are retained:

(i) SLP must increase by $200\,\mathrm{Pa}$ over a distance of $5.5°$ great-circle-distance (GCD) from the candidate point;

   (ii) $Z_{300-500}$ – the geopotential thickness between $300\,\mathrm{hPa}$ and $500\,\mathrm{hPa}$ – must decrease by $58.8\,\mathrm{m}^2\,\mathrm{s}^{-2}$ over a distance of $6.5°$ GCD, using the maximum value of $Z_{300-500}$ within $1°$ GCD of the SLP minimum as a reference.

Criterion (i) ensures that the low-pressure region is of sufficient magnitude and coherent. Criterion (ii) verifies that there is an upper-level warm core associated with the local depression. Finally, candidates for which a stronger SLP minimum exists
within $6°$ GCD are eliminated.

**Stitching TC tracks:** Consecutive candidates are linked together if they lie within $8°$ GCD of one another. A maximum $24\,\mathrm{h}$ gap is allowed in a track, and tracks must last for at least $54\,\mathrm{h}$. Ten 6-hourly time steps ($54\,\mathrm{h}$) must also verify the following additional thresholds: $u_{10} \geq 10\,\mathrm{m\,s^{-1}}$, $|\phi| \leq 50°$, $z_{surf} \leq 150\,\mathrm{m}$, where $\phi$ and $z$ respectively stand for the latitude and the altitude. They respectively ensure that the track is of sufficient intensity, located close enough to the equator, and spends a
significant fraction of its lifetime over oceans.

### 2.3.3 OWZ algorithm

The OWZ algorithm, presented in Tory et al. (2013d) and assessed on ERA-Interim data by Bell et al. (2018) is based on evaluating the eponymous Obuko-Weiss-Zeta (OWZ) quantity, defined according to

$$\mathrm{OWZ} = \max(\mathrm{OW_{norm}}, 0) \times \eta \times \mathrm{sign}(f), \tag{1}$$

where $\eta$ is the absolute vorticity, the sum of the relative vorticity $\zeta$ and the coriolis parameter $f$, and $OW_{\mathrm{norm}}$ stands for the normalized Obuko-Weiss parameter:

$$\mathrm{OW_{norm}} = \frac{\zeta^2 - (E^2 + F^2)}{\zeta^2}, \tag{2}$$

in which $E$ and $F$ are the stretching deformation and the shearing deformation and are given by

$$E = \frac{\partial u}{\partial x} - \frac{\partial v}{\partial y}, \ F = \frac{\partial v}{\partial x} + \frac{\partial u}{\partial y}.$$




**Candidate detection:** Our implementation of OWZ in Tempest Extremes first identifies local maxima of OWZ at $850\,\mathrm{hPa}$. Candidates for which a stronger OWZ maximum exists within 5° GCD are eliminated. Next, only those candidates that satisfy the following six conditions within a distance of 2° GCD of that maximum are retained (with $r$ and $q$ being the relative and specific humidity, respectively, and vws stands for the vertical wind shear between 200 and 850 hPa):

$$\begin{cases} \mathrm{OWZ}_{850\,\mathrm{hPa}} \geq 5 \times 10^{-5}\,\mathrm{s}^{-1} \\ \mathrm{OWZ}_{500\,\mathrm{hPa}} \geq 4 \times 10^{-5}\,\mathrm{s}^{-1} \\ \quad r_{950\,\mathrm{hPa}} \geq 70\,\% \\ \quad r_{700\,\mathrm{hPa}} \geq 50\,\% \\ \quad q_{950\,\mathrm{hPa}} \geq 10\,\mathrm{g\,kg}^{-1} \\ \qquad \mathrm{vws} \leq 25\,\mathrm{m\,s}^{-1}. \end{cases}$$

**Stitching TC tracks:** Consecutive TC points are stitched together when they lie within a maximum distance of 5° GCD from one another, allowing for a maximum 24h gap. Additional core thresholds must be reached for at least 9 time-steps (48h):

$$\begin{cases} \mathrm{OWZ}_{850\,\mathrm{hPa}} \geq 6 \times 10^{-5}\,\mathrm{s}^{-1} \\ \mathrm{OWZ}_{500\,\mathrm{hPa}} \geq 5 \times 10^{-5}\,\mathrm{s}^{-1} \\ \quad r_{950\,\mathrm{hPa}} \geq 85\,\% \\ \quad r_{700\,\mathrm{hPa}} \geq 70\,\% \\ \quad q_{950\,\mathrm{hPa}} \geq 14\,\mathrm{g\,kg}^{-1} \\ \qquad \mathrm{vws} \leq 12.5\,\mathrm{m\,s}^{-1}. \end{cases}$$

Finally, tracks that do not reach tropical storm intensity ($u_{10} = 16\,\mathrm{m\,s}^{-1}$) for at least one time step are filtered out.

Due to the specifics of the Tempest Extremes framework, we note that our implementation differs slightly from the original algorithm described by Tory et al. (2013d). These modifications, along with the results of a sensitivity study justifying our choices for $r_{\mathrm{threshold}}$ and $r_{\mathrm{range}}$ are further discussed in appendix C.

### 2.3.4 TRACK algorithm

TRACK derives from an extra-tropical cyclone tracking algorithm (Hodges, 1994). It is versatile and has since been used to study many types of weather systems, including the detection and tracking of TCs (Bengtsson et al., 2007; Hodges et al., 2017; Roberts et al., 2020a). The rationale behind TRACK is different from the previously described trackers: Because it aims at tracking all vorticity perturbations, it does not embed any warm-core criterion in its initial fundamental detection. The TC selection, including the warm core test, is only performed in the last step independently of the tracking. In the present paper, we used without any modification the database of trajectories detected by TRACK in ERA5 that was recently published by Roberts et al. (2020a). For completeness, we detail below the thresholds used in that case.



The algorithm is based on $\zeta_{T63}(P)$ which is the relative vorticity at pressure level P, spectrally filtered to retain total wavenumbers 6-63 only, as well as its vertical average from $850\,\mathrm{hPa}$ to $600\,\mathrm{hPa}$, hereafter refered to as $\overline{\zeta}_{T63}$. Local extrema of $\overline{\zeta}_{T63}$ are detected and define a series of candidate points provided $\overline{\zeta}_{T63} > 5 \times 10^{-6}\,\mathrm{s}^{-1}$. Neighboring candidates are then stitched together by minimizing a cost function for track smoothness (Hodges, 1995, 1999). The tracks so obtained must last for at least two days and start between 30°S and 30°N.

The presence of a warm core is diagnosed according to the following criteria that must be satisfied for at least one day over the ocean:

(i) $\zeta_{T63}(850\,\mathrm{hPa}) > 6 \times 10^{-5}\,\mathrm{s}^{-1}$;

(ii) $\zeta_{T63}(850\,\mathrm{hPa}) - \zeta_{T63}(250\,\mathrm{hPa}) > 6 \times 10^{-5}\,\mathrm{s}^{-1}$;

(iii) A local maximum of $\zeta_{T63}(P)$ exists at each pressure level.

### 2.3.5   CNRM algorithm

The CNRM algorithm was developed by Chauvin et al. (2006), and later used in Chauvin et al. (2020) and Cattiaux et al. (2020). It has been calibrated for ERA5 by Dulac et al. (in preparation), and this is the dataset we use here. Candidate points are first tracked with the following criteria:

(i) The SLP displays a local minimum which defines the center of the system;

(ii) The 850 hPa relative vorticity is larger than $1.5 \times 10^{-4}\,\mathrm{s}^{-1}$;

(iii) The 850 hPa wind intensity is larger than $5\,\mathrm{m\,s}^{-1}$;

(iv) The sum of the temperature anomalies averaged over the 700, 500, 300 hPa pressure levels is larger than $1\,\mathrm{K}$;

(v) The difference between the 850 hPa and the 300 hPa temperature anomalies is smaller than $1\,\mathrm{K}$;

(vi) The difference between the 300 hPa and 850 hPa wind intensity is smaller than $5\,\mathrm{m\,s}^{-1}$.

This detection step is followed by a stitching procedure adapted from Hodges (1994) and detailed in Ayrault (1998). Tracks shorter than one day are eliminated. Once TC tracks are obtained, a relaxation step is performed to complete the track lifecycle and to detect tracks that were cut in two or more pieces (for example, because of a temporary weakening). This relaxation step is done with a 850 hPa relative vorticity threshold equal to $2.5 \times 10^{-4}\,\mathrm{s}^{-1}$.

### 2.4   Tracks matching

When using reanalysis products like ERA5, detected tracks can tentatively be associated with observed tracks (Murakami, 2014; Hodges et al., 2017; Ullrich et al., 2021). We used the following matching algorithm. Consider the case of a given detected track D composed of n points $(d_1, d_2, ...d_n)$ defined at times $(t_1, t_2, ..., t_n)$. The observations O consist of a database of tracks and can be seen as a collection of points at given times. For each point $d_i(t_i)$ of the track D, we associated those points of O at time $t_i$ that are located closer than 300 km from the point $d_i$. Of course, it is possible that such points do not exist

in O. The subset of points of O that have been associated with any point in D is noted $O_{\mathrm{D-paired}}$. It is composed of $|O_{\mathrm{D-paired}}|$ elements. There are three possibilities:

(i) $|O_{\mathrm{D-paired}}| = 0$: None of the points of D has been paired to a point in O. D is considered to be a false alarm.





(ii) $|O_{\text{D–paired}}| > 0$ and all the points in $O_{\text{D–paired}}$ belong to the same track $D_O$ in O: $D_O$ is considered to be the match of D.

(iii) $|O_{\text{D–paired}}| > 0$ and the points in $O_{\text{D–paired}}$ belong to more than one track in O: the observed track having the largest number of points paired with D is considered the match of D.

After this matching is completed for all detected tracks, a final treatment is performed: if an observed track is paired with two or more detected tracks, these detected tracks are merged into a single track. Such cases arise when the detected track corresponds to different parts of the same observed tracks and occur when, for example, the TC temporarily weakened while going over an island before strengthening again. In appendix D, we present a rapid analysis that validates our method.

This matching procedure enables us to label tracks as "Hits" (H), "Misses" (M), and "False Alarms" (FAs). Hits are tracks present in IB–TS and detected in ERA5. Misses are tracks present in IB–TS that were not detected in ERA5. False Alarms are tracks detected in ERA5 that do not correspond to any track in IB–TS. We then used this labeling to define two detection skills metrics, the Probability of Detection (POD, sometimes also presented as HR for Hit Rate) and the False Alarm Rate (FAR):

$$POD \quad = \quad \frac{H}{H + M}, \tag{5}$$

$$FAR \quad = \quad \frac{FA}{H + FA}. \tag{6}$$

## 3  A common trackers post-treatment

We used Eq. (5) and (6) to calculate the POD and FAR of the four trackers with respect to IB–TS. For UZ, we found a POD of $75\%$ and a FAR equal to $18\%$. These values are almost identical to Zarzycki et al. (2021), who report $78\%$ and $14\%$ for their POD and FAR, respectively. Subtle differences in the pre-processing of the IBTrACS data account for this difference (Zarzycki, private communication) but the fact that both PODs and FARs are almost identical validates our implementation of that tracker. For TRACK, we found a POD of $85\%$ and a FAR equal to $50\%$. Both scores are comparable to the values reported by Hodges et al. (2017) who applied TRACK to other reanalyses. We note that the POD we report here is on the higher end of the values found by Hodges et al. (2017), which is consistent with our more restrictive filtering of IBTrACS than Hodges et al. (2017). OWZ and the CNRM tracker display PODs similar to UZ, but their FARs are more heterogeneous and amount to $28\%$ for OWZ and $60\%$ for the CNRM tracker.

Overall, the results demonstrate that all trackers can capture most of the observed TCs. Although this is satisfying, we note that a given tracker can miss up to one-fourth of the existing tracks. In addition, as stated above, the FARs are more heterogeneous, and FAs can account for more than half of the detected trajectories. These two caveats call for a better understanding of the properties of both populations. This is the purpose of the following section.

### 3.1  Missing tracks and False Alarms properties

Figure 1 reports several diagnostics that characterize the different populations (hits, misses, and false alarms) detected in ERA5 by the trackers.





**Figure 1.** Histograms representing the properties of the Hits, the Miss, and the False Alarms tracks for each tracking algorithm. From left to right, the columns correspond to UZ, OWZ, TRACK, and the CNRM tracker, respectively. The rows correspond from top to bottom to the minimum sea-level pressure (with the storm categories as defined according to Table 1 shown with vertical gray lines), the latitude at which that value is reached, the month at which that value is reached (solid line in the northern hemisphere, and dashed line in the southern hemisphere), and finally the track duration. The blue and green colors correspond to the Hits and the Misses, respectively, for all plots. Raw False Alarms are shown in orange while we plot in red the False Alarms that remain after the post-treatment (see section 3 for details).





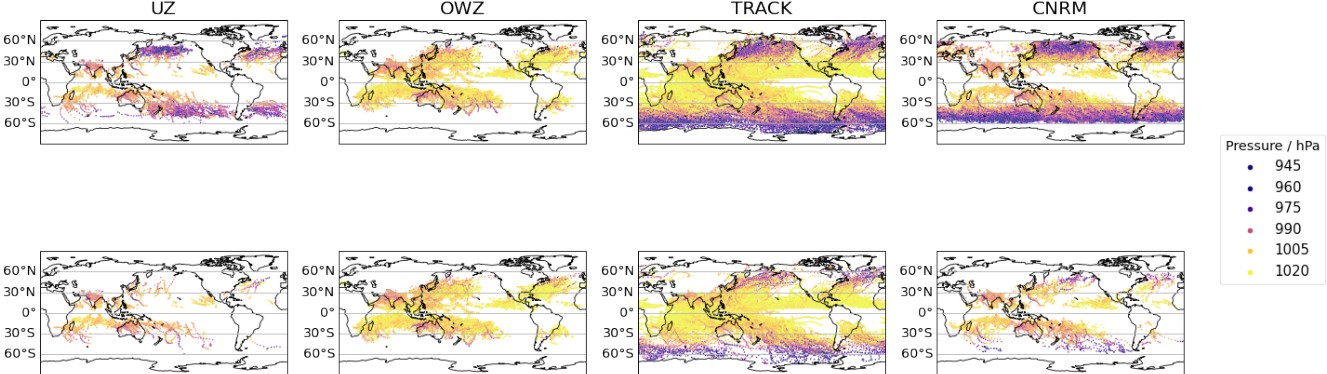

**Figure 2.** Top row: maps of the FA tracks color-coded according to their intensity in terms of pressure. The different columns each correspond to a different tracker. Bottom row: same as the top row, but after the STJ post-treatment has been applied (see section 3).

For practical purposes, we use the hits (blue color in Fig.1) as a reference against which to compare these diagnostics. TCs intensity distribution (first row) and seasonal cycle (third row) are similar across all trackers. The seasonal cycle is the same as in the observations, but the intensity distribution is under-estimated (see section 4.4). We find some differences in the latitude at which the SLP minima are reached (second row). The CNRM tracker distribution features secondary maxima in midlatitudes in both hemispheres that are absent in UZ and OWZ and only barely visible in TRACK (particularly in the southern hemisphere). These tracks correspond to TCs that reached their maximum intensities after a post-tropical transition. Hits lifetimes (fourth row) also vary with trackers: UZ and the CNRM display the shortest tracks with a distribution that peaks between 5 and 10 days, followed by OWZ with storms durations peaking at 10 days, while TRACK tracks typically last for 15 days. We will come back in more detail to these properties in section 4.3.

Missing tracks (green color in Fig.1) correspond to tropical storms or cyclones that were observed and are reported in the IB–TS database but that a given tracker did not find in ERA5. They typically consist of weak (first row), tropical (second row), and short-lived (fourth row) perturbations for which the amplitude is probably not strong enough to exceed the detection thresholds for long enough time[2]. This is why TRACK, with its relatively soft criteria, misses half as many tracks as the other trackers. We also note that the latitudinal distribution of missed tracked (second row) is skewed in favor of the northern hemisphere, a property they share with the population of hits. Because they are observed as tropical storms, missing tracks are more numerous during the TC season of their hemisphere (third row). To conclude, the missed trajectories seem to correspond to the weak tail of the distribution of hit trajectories. Our description of missing tracks is in agreement with Hodges et al. (2017).

FAs (orange color in Fig.1) correspond to perturbations detected in ERA5 by a given tracker for which there exists no correspondence in IB–TS. FA storms are not systematically weak and their intensity distributions vary across trackers (first

---

[2]Some care is required here: by definition, the properties of missing tracks reported in Fig.1 rely on the information contained in IBTrACS, while that of hits comes from ERA5. While this is probably not a concern for the latitudes of pressure minimum and for the track duration, this may be more problematic for the pressure minimum itself, as modeled TCs tend to reach weaker intensities than observed TCs





row). The CNRM tracker shows the most extrem distribution of FAs, with a peak that corresponds to category 2 storms. By contrast, OWZ FAs distribution is strongly biased toward weak category 0 storms. UZ and TRACK FAs strength distributions simultaneously show weak storms along with a significant tail of strong storms – in the sense that the number of category

1 and 2 storms is not negligible compared to the number of category 0 storms. The second row of figure 1 suggests that these relatively strong disturbances correspond to extra-tropical cyclones. Indeed, the latitude distribution of the minimum SLP value shows two peaks at mid-latitudes, for UZ, TRACK and the CNRM tracker. For the latter, these peaks even exceed the subtropical peaks associated with the hits. In agreement with that hypothesis, the seasonality of the FAs in UZ, TRACK, and CNRM shows that there is an important number of storms detected during the winter season of each hemisphere, i.e.,

precisely when extra-tropical cyclones are numerous (figure 1, third row). By contrast, OWZ FAs occur during the TC season. For all trackers, the ratio of summer to winter false alarms is consistent with the ratio of the peaks observed at tropical and midlatitudes in the latitudinal distribution of FAs: UZ and TRACK FAs have rather flat seasonal cycles, and the same number of tropical and extra-tropical FAs, the CNRM tracker has most of its FAs during winter at extra-tropical latitudes, and OWZ FAs mainly occur during the TC season at tropical latitudes. Finally, FAs are generally shorter events than hits (last row). UZ

and CNRM FAs tracks are the shortest and last less than 10 days. TRACK FAs tracks are the longest and feature durations of up to a month. OWZ FAs can last up to 20 days. Interestingly, Bell et al. (2018) also reported similarly long FAs while tracking TCs in ERA-Interim with OWZ. They were then able to associate the longest FAs with observed tropical disturbances that had been discarded from IBTrACS because they only retained storms of tropical intensity and stronger, as we did in this paper. Although we did not do the same exercise, in light of their results, it is likely that some of the FAs we report here also

correspond to weak storms we excluded from IBTrACS.

We conclude that FAs belong to two categories: (i) strong extra-tropical (ET) and (ii) weak tropical storms. This conclusion is nicely illustrated and confirmed with the help of FA tracks maps (figure 2, top row): For UZ and the CRNM tracker, FA tracks correspond to intense storms (pink colors) that cluster beyond 30° latitude. On the other hand, OWZ FA tracks are tropical and weak disturbances (yellow colors). TRACK FA tracks are of both types: many of them are strong ET storms, but there is also

a large contingent of weak tracks.

### 3.2    Post-treatment: two methods

The discussion above identified two types of FAs: weak, short-lived tropical storms and strong extra-tropical cyclones. It seems complicated to filter the weak and short-lived tracks because such a procedure would simultaneously remove many hits and significantly reduce the POD. For example, $24\,\%$ to $83\,\%$ of the tracks (for TRACK and UZ, respectively) with a minimum

pressure larger than $1005\,\mathrm{hPa}$ are hits.

By contrast, extra-tropical cyclones are sufficiently different from genuine TCs to derive a discriminating method. We note that such an avenue for improvement has already been explored in the past. For example, based on the fact that extra-tropical cyclones preferentially develop in midlatitudes, some trackers use a fixed latitude criterion to filter out some of the tracks suspected to correspond to extra-tropical cyclones (see e.g. Table 1 in Chauvin et al., 2006). Such a simple criterion may not

be elaborate enough, though. For example, it does not take into account the natural variability of the sub-tropical limit nor its





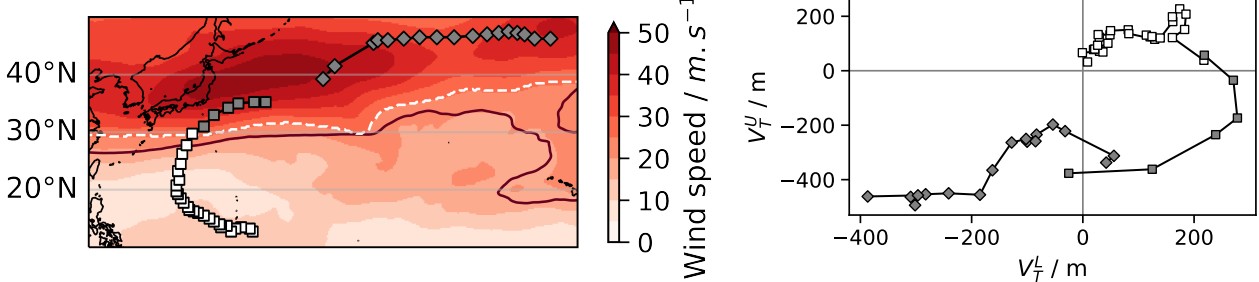

**Figure 3.** Illustration of the post-treatment on two simultaneous tracks: One that both post-treatment methods have discarded, and one that corresponds to typhoon Mac (observed cat. 5, reanalyzed cat. 3). Different markers differentiate them. The left panel shows the tracks in physical space, the STJ limit (dashed white line), overlaid with the horizontal wind speed (red shading) and the zonal wind speed (red contour), both at 200 hPa. The right panel shows the track in the Hart phase space. In both plots, grey points correspond to points that have been labeled ET by the STJ post-treatment.

potential poleward shift with climate change. In fact, the two trackers in this study that embed such a cut-off parameter (UZ and TRACK) still present a large number of ET tracks, suggesting there is room for improvement. An alternative option is to rely on the structural differences between tropical and extra-tropical cyclones, for example, the nature - warm or cold - of their core.

In the following, we develop and analyze the results and relative merits of both approaches. We propose two post-treatment methods inspired by the existing literature: (1) an adaptation of Bell et al. (2018) SubTropical Jet (STJ) cut-off, hereafter called the STJ method, and (2) an exploitation of Hart phase space diagram (Hart, 2003), hereafter called the VTU method.

**The STJ method** (see fig. 3, left panel for a graphical illustration) is an environmental method that aims at establishing an objective criterion to determine whether a given disturbance is located in the midlatitudes or the tropics. It is based on the

large-scale wind field properties at $200 \, \text{hPa}$. First, we apply a 30-days running mean on both wind components to remove the fast atmospheric synoptic activity. The STJ is then defined as the region where the wind speed $\sqrt{u_{200}^2 + v_{200}^2}$ is larger than $25 \, \text{m s}^{-1}$ and the zonal wind $u_{200}$ is larger than $15 \, \text{m s}^{-1}$. At each time-step, we define the maximum tropical latitude for each longitude as the equatorward boundary of the STJ. For those longitudes where no STJ exists, the boundary latitude is linearly interpolated between the two closest longitudes with an existing STJ. Any disturbance located poleward of that limit

is assigned an extra-tropical label. We eventually filter out tracks that feature no or only one tropical point.

**The VTU method** (see fig. 3, right panel for a graphical illustration) is a structural method that aims at establishing an objective criterion to discriminate between TCs and ETCs. Here we use the Hart phase space diagram that plots storms trajectories in a 2D diagram based on measures of the storm thermal wind in the upper and lower troposphere, respectively noted $V_T^U$ and $V_T^L$ (Hart, 2003). Here, we used the following relation to calculate $V_T^U$:

$$V_T^U = P_{mid} \frac{\Delta Z(P_{bottom}) - \Delta Z(P_{top})}{\Delta P},$$





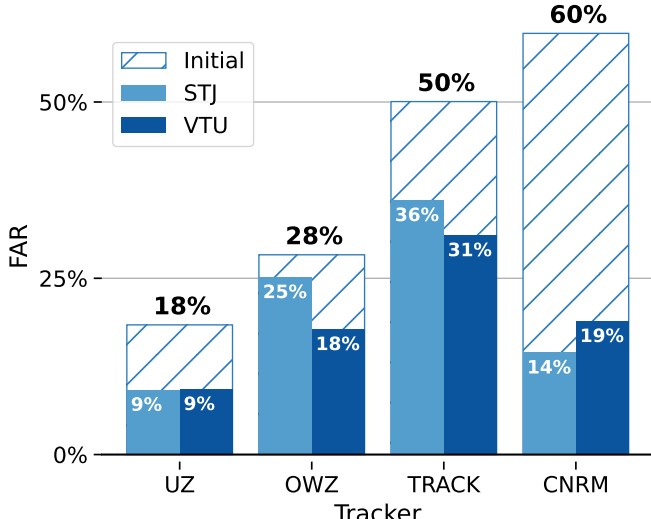

**Figure 4.** FA rates for each algorithm, before (hatched, black figures) and after (filled, white figures) post-treatment.

where $P_{top} = 300\,\mathrm{hPa}$, $P_{bottom} = 600\,\mathrm{hPa}$, $\Delta P = P_{top} - P_{bottom}$ and $P_{mid} = (P_{top} + P_{bottom})/2$. $\Delta Z(P)$ is the maximum height perturbation on the isobaric surface of pressure $P$ within a circle of 500 km radius centered on the storm:

$$\Delta Z(P) = Z_{max}(P) - Z_{min}(P).$$

$V_T^L$ has a similar definition but with $P_{top} = 600\,\mathrm{hPa}$ and $P_{bottom} = 900\,\mathrm{hPa}$. As noted by Hart (2003), storms trajectories in
the $(V_T^L, V_T^U)$ plane are enlightening as to the nature of the storm, and we have found that $V_T^U$ is a powerful discriminant between full-troposphere warm-core TCs and other structures. In practice, the VTU method consists in filtering out tracks for which $V_T^U$ is negative for all time-steps.

### 3.3 Post-treatment: the results

Both post-treatment schemes are efficient at reducing the FARs (see fig. 4). The STJ method removes between $11\,\%$ and $76\,\%$
(for OWZ and CNRM, respectively) of the FAs, while the corresponding reductions range from $37\,\%$ and up to $68\,\%$ (for OWZ and CNRM, respectively) with the VTU method. The STJ reductions in FAR correspond to the proportion of extra-tropical FAs identified in section 3.1: Barely any in OWZ, about half in UZ and TRACK, and most of CNRM tracks.

In figure 1, we further compare the properties of the FAs before and after the STJ post-treatment, respectively with the orange and red distributions (The effects of the VTU method on the FAs distributions are shown in Fig. B2 and are almost
identical). The large amplitudes of the tail of strong storms and the secondary peaks at mid-latitudes are significantly reduced for all trackers (first and second row). Both distributions are now similar to that of the hits. Likewise, the seasonal cycle of the filtered FAs looks more similar to that displayed by actual TCs (third row). Visual inspection of STJ–filtered FA tracks (fig. 2,





second row) confirms these quantitative diagnostics and shows that the filtering procedure has dramatically reduced ET tracks frequencies. We conclude that the STJ method fulfills its goal of selectively removing ETC tracks

The FARs after post-treatment are similar with both methods. They range from $9\%$ up to $36\%$ for the STJ method, and from $9\%$ up to $31\%$ for the VTU method (see also Table 2). OWZ seems to be an exception to that rule, though: while the STJ method leaves its FAR nearly unchanged, the VTU method succeeds in removing more than one-third of its FAs. This relatively poor performance of the STJ method at removing OWZ FAs was to be expected: As discussed above in section 3.1, ET storms do not dominate its population of FAs, which rather appear to be composed mostly of weak short storms. This is

most likely because OWZ already embeds a wind shear criterion in its formulation. It probably already detects the crossing of the sub-tropical jet, thereby reducing the interest of the STJ filtering method. By contrast, the better performance of the VTU method for that tracker suggests that it is more efficient at identifying weak/short FA tracks and makes it more interesting to use in combination with OWZ. Nevertheless, we note that our results are in agreement with Bell et al. (2018) who report a decrease of 2.5 and 4.5% of the total tracks in the NH and SH, respectively, when they used an STJ-like criterion on ERA-Interim data.

In our case, we found that the STJ post–treatment removes 4% of all the OWZ tracks. The detection scores obtained for OWZ after the STJ post-treatment are also close to those obtained by Bell et al. (2018) with ERA-Interim, i.e., a 73% POD and 19% FAR.

As mentioned above, a desired property of any post-treatment procedure is to leave the POD unaltered. We found that the two methods display some differences (see fig. 5). While the STJ method only reduces the POD by $1\%$ at most for all trackers,

the VTU method has a larger impact: PODs decrease from $3\%$ (for TRACK) to $7\%$ (for the CNRM tracker). The VTU post-treatment even removes up to $4\%$ of TC-strength hits in UZ and CNRM. For this reason, we only present results obtained using the STJ method in the remaining of this paper. It does not mean that the VTU post-treatment should always be discarded: as opposed to the STJ method, it only requires information about the local and instantaneous properties of the flow. It is thus simpler to implement than the STJ method. This relative simplicity has a price to pay in terms of a modest decrease of the

PODs that one should be aware of.

In addition to filtering out ETCs, the post-treatment methods described above allow us to label ET points in the remaining tracks. These ET points are then excluded when computing the intensity statistics of the tracks (see section 4). This "free bonus" of the post-treatment step removes potential biases in the metrics that would result from TC tracks that reach their maximum intensity after performing a post-tropical transition.

**4 Results**

We now analyze the properties of the database of ERA5 tracks we obtained after the post–treatment described above, focusing on the differences between trackers. We first come back to the trackers detection skills (section 4.1). We then discuss the sensitivity of the metrics introduced by Zarzycki et al. (2021) in section 4.2 and then relate these sensitivities to the different tracks' duration as captured by the four trackers (section 4.3) and to the intensity distribution of the reanalyzed storms in ERA5

(section 4.4).





| Tracker | UZ | OWZ | TRACK | CNRM |
|---------|-----|-----|-------|------|
| POD | 74% | 76% | 84% | 74% |
| FAR | 9% | 25% | 36% | 15% |

**Table 2.** Probability Of Detection (POD) and False Alarm Rate (FAR) of the four trackers used in this work with respect to IB–TS.

## 4.1 Trackers detection skills

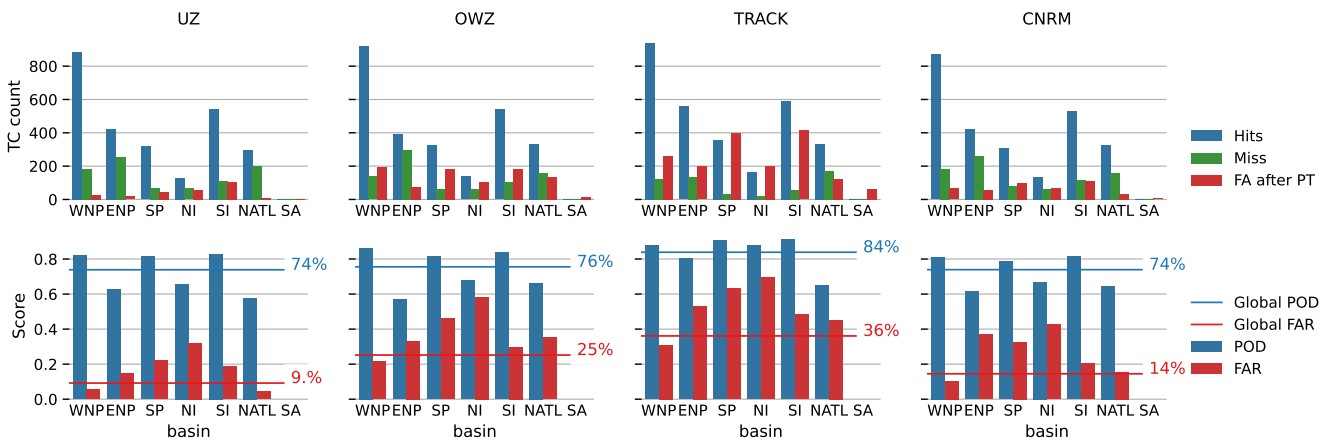

**Figure 5.** Upper panel: Hits, misses and FA total numbers per oceanic basins. From left to right, the different panels correspond to UZ, OWZ, TRACK and the CNRM tracker, respectively.

Lower panel: POD and FAR for each tracker in each basin (bars) compared to the global mean (lines).

Basins acronyms are defined as follows: Western North Pacific (WNP), Eastern North Pacific (ENP), South Pacific (SP), North Indian (NI), South Indian (SI), North ATLantic (NATL) and South Atlantic (SA).

Table 2 summarizes for completeness the filtered trackers detection skills extensively discussed in the previous section. The PODs are almost unchanged compared to the values discussed in section 3 before post-treatment, and the FARs are as shown in figure 4 for the STJ method. Overall, these numbers illustrate the trade-off between false alarms and missing: improvements of the POD tend to occur at the cost of an increase in the FAR.

However, these numbers are global averages and hide a significant regional variability. This variability is illustrated in figure 5 (top row), which decomposes the numbers of hits, misses, and FAs by oceanic basins[3]. First, we note that the hits' geographical distribution is similar across trackers: they are more numerous in the western north Pacific ocean, followed by the Southern Indian ocean, the Eastern North Pacific and finally the South Pacific and North Atlantic which features almost the same number of TCs. The geographical distribution of misses is not identical to that of the hits and varies among trackers. This variability translates into POD values than can strongly deviate from the mean (figure 5, bottom row). For example, the

---

[3]Here and in the remainder of the paper, oceanic basins are defined following Knutson et al. (2020) appendix guidelines.



| $year^{-1}$ | Frequency # | TC days days | ACE 1e4 $(m/s)^2$ | $\phi_{P_{min}}$ ° lat |
|---|---|---|---|---|
| Obs | 88 | 776 | 168 | 20 |
| UZ | -16 | -334 | -107 | 0.6 |
| OWZ | 1 | 50 | -91 | 0.9 |
| TRACK | 27 | 731 | -92 | -2.3 |
| CNRM | -12 | -310 | -106 | 0.9 |
| Mean bias | 0 | 34 | -100 | 0 |
| $\sigma$ | 20 | 496 | 8 | 1.6 |

**Table 3.** Frequency, TC days, ACE and latitude of minimum pressure ($\phi_{P_{min}}$) in the observations, and bias in ERA5 depending on the tracker used. The last two lines show the mean and the standard deviation of the bias with regard to the trackers.

POD in the North Atlantic is smaller than the global average by $10\%$ for all trackers and only reaches $58\%$ for UZ. Misses are also more numerous in the Eastern North Pacific, although with contrasted results among trackers: while UZ, OWZ, and CNRM PODs roughly equal $60\%$, it amounts to $80\%$ for TRACK, i.e., close to its global average. We find similar figures for the Northern Indian ocean. These problems are balanced by POD scores that are systematically larger than the global averages for the Western North Pacific (WNP), South Pacific, and South Indian (SI) oceans, where the PODs are larger than $80\%$. With almost two-thirds of the world's TCs occurring in the WNP and the SI oceans, these two basins largely account for the global averages reported in Table 2.

Similarly, the FAs geographical distribution does not necessarily follow that of the hits and is heavily weighted by the WNP value. In this basin, the FAR is equal to $8\%$ and $10\%$ for UZ and the CNRM trackers, respectively, and largely explains the low FARs for these two trackers. It amounts to $20\%$ and $30\%$ for OWZ and TRACK, also reflecting their global average values. In many of the other basins, FARs are much worse than their global average and often exceed $40\%$. This is particularly true for the southern oceanic basins and the Northern Indian ocean. In fact, regional FARs smaller than the global mean are an exception rather than the rule.

## 4.2 Metrics sensitivity

We now take the different point of view of assessing the properties of the detected tracks as an ensemble composed of the hits and the false alarms aggregated together. We compare them with the properties of the observed tracks as derived from IBTrACS. Such an approach would be more appropriate when using trackers to evaluate model results as opposed to reanalysis for which detection scores can be calculated.

To do so, Zarzycki et al. (2021) suggested to use a series of standard metrics as a means to evaluate the performance of a system in simulating tropical storms against an observed reference. Using the UZ tracker (albeit without the post-treatment method described above), they applied it to several reanalysis products. Here, we take a complementary viewpoint and use a

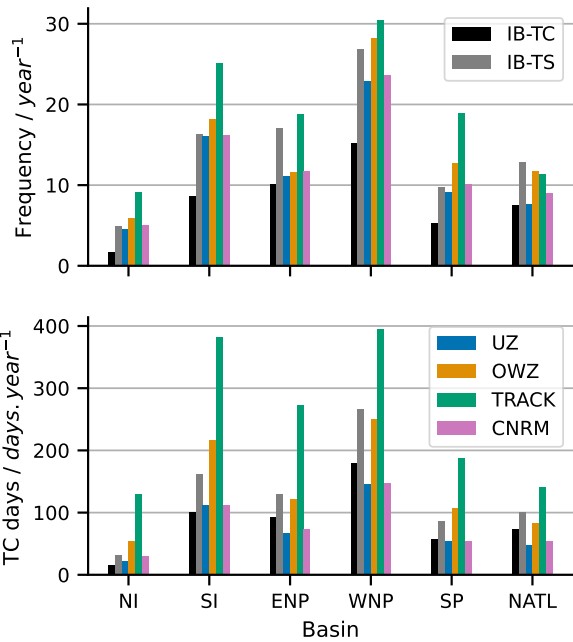

**Figure 6.** Regional distribution of frequencies top panel) and TC days (bottom panel) for each tracker and in the observations. Basins acronyms are as defined in figure 5.

subset of their metrics to evaluate the performances of several trackers against a single reanalysis product. Table 3 reports the bias we measured for each of these metrics and for each trackers with respect to IB–TS.

The global storm frequencies – i.e., the total number of storms detected per year – vary among trackers and reflect their different sensitivities: UZ and the CNRM tracker are the most selective and display a negative bias. TRACK is the most sensitive tracker and has a positive bias. OWZ behavior is intermediate and features a very small bias. Maybe more important than their absolute value is the fact that the biases standard deviation ($\sigma = 20$ yr$^{-1}$) amounts to more than $20\%$ of the observed track frequency. It is also comparable to the dispersion of track frequencies of $18.3$ yr$^{-1}$ reported by Zarzycki et al. (2021) in

their analysis of a series of reanalysis products with a single tracker. This comparison indicates that uncertainties associated with using a single reanalysis product are as large as those associated with using a single tracker.

Zarzycki and Ullrich (2017) and Zarzycki et al. (2021) advocate using more integrated metrics, such as the total number of days featuring TCs, also referred to as TC days. We recover bias of the same sign as for the frequencies for that metric. For UZ, the bias is very different in both its sign and amplitude from the value reported by Zarzycki et al. (2021). This is because

we consider here the entire trajectories reported in IBTrACS, while Zarzycki et al. (2021) only included storms of tropical storm strength (i.e. with $u_{10} > 16$ m.s$^{-1}$). The observed TC days they calculated is thus smaller than the values we report in Table 3, explaining the differences between the biases. Even if the number of TC days is an integrated metric, its scatter among the different trackers is even larger than found for the frequencies and amounts to $63\%$ of the IB–TS value. This large scatter



is due to the fact that TC days multiplies TC frequencies with track duration. As already discussed in section 3.1, the latter is
variable among trackers, and that variability is positively correlated with trackers' sensitivities: tracks durations increase for
sensitive trackers. We will come back to that aspect in section 4.3.

As already mentioned in section 4.1, there is significant regional variability of the POD and the FAR. This is also the
case for the aggregated catalog (hits plus FAs), as illustrated in figure 6 (top row), where we also compare our results with
both IB–TS and IB–TC. First, we note that TCs' frequency biases with respect to IB–TC are positive for all trackers and all
basins. The comparison with IB–TS is more variable. We recover the negative biases in frequencies of UZ and CNRM for all
basins, although with different amplitudes: it is large in the Eastern North Pacific but nearly vanishes in the South Indian and
South Pacific oceans. Likewise, OWZ features smaller biases but with different signs depending on the basins and occasionally
displays large values, for example, for the Eastern North Pacific. TRACK biases also tend to be positive and large, in line with
the global positive bias, except for the North Atlantic. The low POD we already noticed in that basin is not compensated by the
FAR, and the number of detected tracks remains smaller than observed even for that sensitive tracker. Surprisingly, this is also
the only basin where OWZ outnumbers TRACK. Concerning TC days, the geographical distribution (figure 6, bottom row)
visually confirms the larger scatter for that metric than for the frequencies. However, the biases with respect to IB–TS appears
to be more homogeneous, with large negative biases obtained for UZ and CNRM, a small bias for OWZ and large positive
biases for TRACK, occasionally showing TC days larger by more than a factor of two compared to the observed value, as is the
case for example in the Southern Indian and Pacific oceans and for the Eastern Northern Pacific. This consistency between the
regional and global biases also manifests itself in the good spatial correlations between the observed and detected catalogues:
in agreement with Zarzycki et al. (2021), we indeed found a correlation coefficient of 0.97 between UZ and IB–TS, while we
obtained similar albeit slightly smaller values of 0.93, 0.85 and 0.96 for OWZ, TRACK and the CNRM, respectively.

Table 3 also reports the values of the Accumulated Cyclone Energy (ACE), which is a measure of the storms maximum
kinetic energy:

$$ACE = 10^{-4}\Sigma u_{10,max}^2 \tag{7}$$

where $u_{10,max}^2$ is the maximum 10 meters wind speed reached by each tracks and the sum is over the total number of detected
or observed tracks. In agreement with Zarzycki et al. (2021), the ACE bias is negative for UZ as well as for the other trackers.
The values are also much more homogeneous because the ACE is heavily weighted by the more powerful TCs for which the
different trackers agree. We will come back to that point in section 4.4.

Finally, the latitudes of minimum pressure $\phi_{P_{min}}$ is well represented in ERA5 (table 3, last column). UZ bias is smaller than
reported by Zarzycki et al. (2021) and the actual value of $\phi_{P_{min}}$ is closer to the observed value. This reduction is a consequence
of the removal of ET tracks by the post-treatment. Before filtering, we indeed found a bias in $\phi_{P_{min}}$ equal to $3.5°$. This is also
in agreement with the interpretation of Hodges et al. (2017) who also found positive biases for a large number of reanalysis
when using TRACK. In our case, we note that TRACK is the only tracker with a negative bias, a fact we attribute also to the
post–treatment, and to the large number of FAs: the latter are mostly composed of short and weak storms that preferentially
develop equatorward of the population of hits.





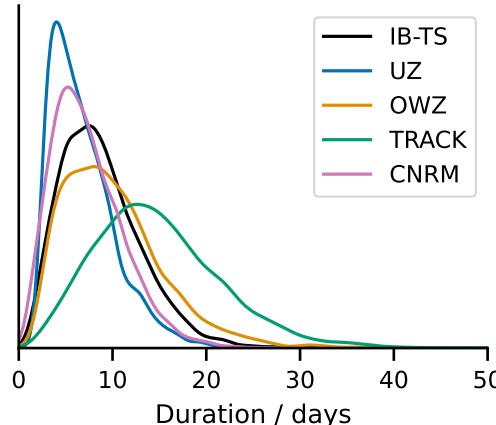

**Figure 7.** Normalized distribution of track duration for all tracks detected by each tracker, after the STJ filtering, compared to IB–TS and IB-TC.

### 4.3 Track duration

In section 4.2, we argued that the increased scatter for TC days compared to that of the frequencies was due to the differing

track durations detected by the different trackers (see also section 3.1, which highlights that issue for the hits). Figure 7 shows that this is indeed the case: track durations are ranked according to the tracker sensitivity, and the corresponding distributions are found to peak at 5, 6, 8, and 12 days for UZ, the CNRM, OWZ, and TRACK, respectively. Hodges et al. (2017) already showed such long TRACK storm durations for other reanalyses products. We find here that they are also longer than IB–TS tracks. OWZ tracks' durations are closest to the observations, while UZ and CNRM tracks are shorter than IB–TS tracks.

To illustrate that point further, we can compare the first and final dates of detected and observed tracks (figure 8). This comparison demonstrates that tracks' durations are homogeneous across oceanic basins. The only exception may be the Western North Pacific, where the results suggest a tendency for UZ and the CNRM tracker to detect tracks later than other oceanic basins. In general, TRACK detects the most extended TC lifecycle: more than 75 % of its tracks start four to five days before the first IBTrACS record and terminate two to three days after the last IBTrACS record. Figure 8 also shows that three-quarters

of OWZ tracks start before IBTrACS by two to three days but present a reduced ability to follow a track after its extra-tropical transition. UZ and CNRM tracks are very similar to IBTrACS ones, although slightly shorter in general. These results may sound surprising at first because they appear to disagree with figure 7 where we found that OWZ tracks correspond to IB–TS while UZ and CNRM tracks were significantly shorter. The difference is due to the FAs: necessarily, figure 8 is restricted to matching tracks, i.e., to the hits. As discussed above, FAs are mainly composed of short and weak tracks, which reduces the

mean duration of the trackers' trajectories, explaining the differences between the two figures. Interestingly, we note that the two dynamics-based trackers in our study share the capacity to detect the storms early in their development.




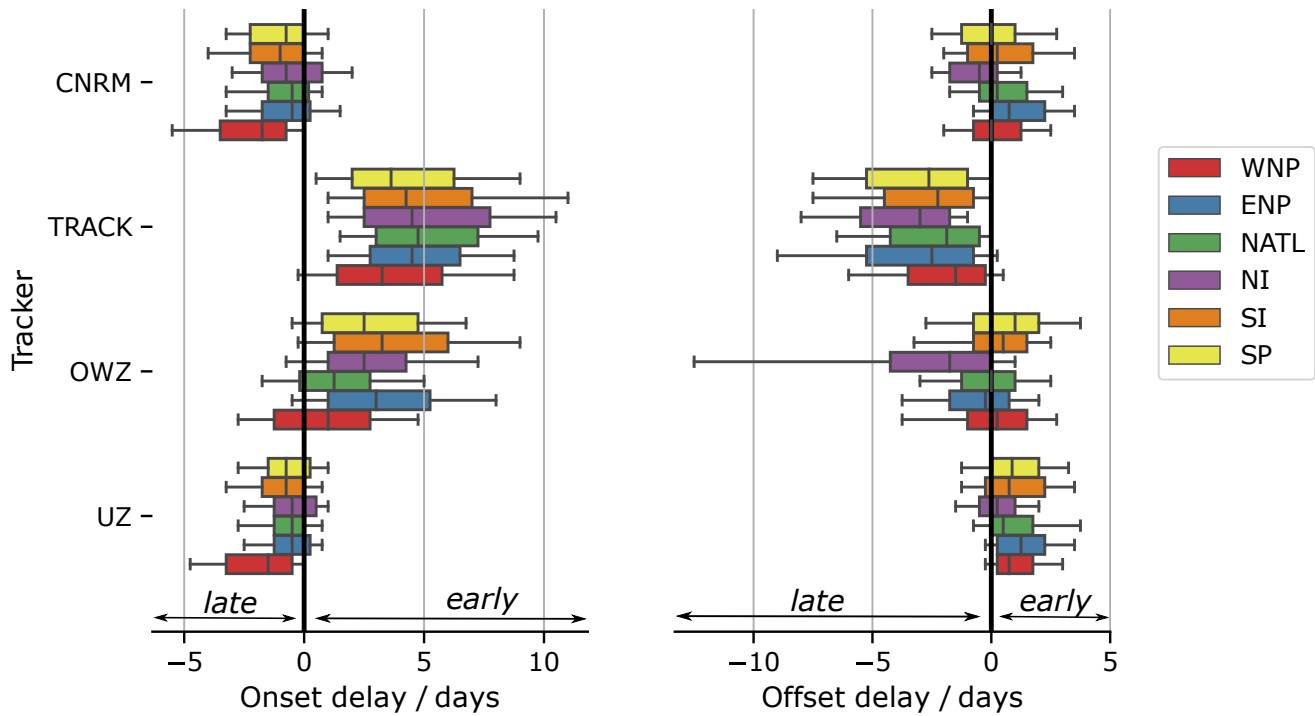

**Figure 8.** Delay (in days) between the detection by each trackers and the record for the corresponding strom in IBTrACS. The different colors correspond to different basins, whose acronyms are as defined in figure 5. Boxplots display 25th, 50th and 75th percentiles, whiskers display 10th and 90th percentiles, outliers are not shown.

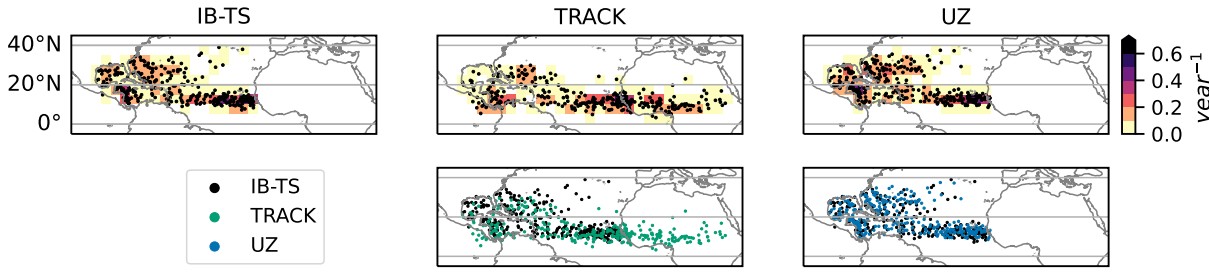

**Figure 9.** First observed/detected points in IB-TS, TRACK and UZ. Top row shows the first points along with the corresponding density. Bottom row overlays IB-TS tracks first point with ERA5 tracks first point as detected tracks by TRACK (second column) and UZ (third colunm).

One can use the ability of TRACK to detect storm tracks early in their life cycle to study TCs genesis locations. For example, in the North Atlantic, TCs are known to originate from Africa and to be mostly triggered by African Easterly Waves (Landsea,





1993; Avila et al., 2000). In IBTrACS, the first reported point of TCs tracks tends to be located in the eastern central Atlantic
ocean, in agreement with the tracks detected in ERA5 by UZ (Figure 9, top and middle panels). By contrast, TRACK genesis
locations extend further east and well over the African continent (Figure 9, bottom panels), i.e., well into the region where
African Easterly waves develop. These differences in genesis locations illustrate TRACK's ability to follow from very early
on the vorticity perturbations that later transform into genuine TCs, and potentially to associate these early perturbations with
known atmospheric phenomena. This property of TRACK opens the way for studying the correlation between North Atlantic
TC genesis and African easterly waves in ERA5, in the spirit of studies such as done by Thorncroft and Hodges (2001);
Hopsch et al. (2007) and Duvel (2021). It would be interesting to further exploit that property by performing similar studies of
TC precursors in other oceanic basins systematically.

## 4.4 Intensity

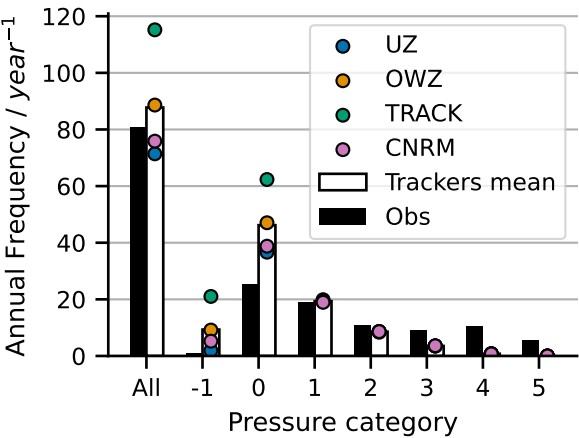

**Figure 10.** Annual frequency depending on the pressure category, in IBTrACS (black bars), and as found by each tracker in ERA5. The
colored dots show the result of each trackers and the mean is represented with the white bar. The -1 category correspond to tracks that did
not reach the $1005\,\mathrm{hPa}$ threshold for category 0.

Section 4.2 reported a large and negative bias of the ACE for all trackers. This is because the intensity distribution of re-
analyzed TCs is different from the intensity distribution of observed TCs (Figure 10). For all trackers, there is a negative bias
for storms of category 2 and larger and an excess of weak storms of categories 0 and -1, which is due to both FAs and hits
reanalyzed in ERA5 with a weaker intensity than observed. The two biases compensate so that the overall TC frequencies are
comparable in ERA5 and IBTrACS. The difficulty of models in general to simulate strong cyclones is well known (Roberts
et al., 2015; Manganello et al., 2012; Strachan et al., 2013; Davis, 2018). It is often illustrated using so–called wind-pressure
diagrams such as shown on figure 11 for UZ (the wind–pressure diagrams obtained using the other trackers are almost indis-
tinguishable). In agreement with the ACE negative bias and with figure 10, figure 11 confirms that detected TCs are weaker



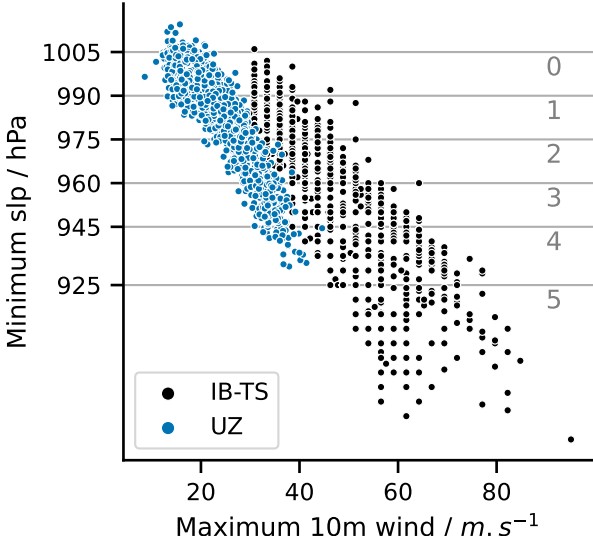

**Figure 11.** Wind-pressure relationship in IB–TS and in ERA5 (UZ tracking).

than observed. They do not follow the same wind-pressure relationship as seen in the observations. In ERA5, TCs maximum wind speeds are even more dramatically reduced than TCs minimum pressure compared to the observations. The problem is not specific to ERA5. It is encountered in all reanalyses, especially in ERA5's predecessor ERA-Interim (Hodges et al., 2017; Schenkel and Hart, 2012; Murakami, 2014; Bell et al., 2018). Zarzycki et al. (2021) report an improvement from ERAI to ERA5 in terms of ACE, but it is obvious from figures 10 and 11 that ERA5 remains heavily biased.

Figure 10 also reveals that all trackers agree very well for intense TCs of category 2 and above, so the following result holds: all of the detected strong cyclones are detected by all trackers, all of the detected strong cyclones are hits (see the red distributions on the first row of figure 1), and there are no FAs among the detected strong cyclones (see the green distributions on the first row of figure 1). The spread between trackers is only due to FAs and misses of weak tropical storms.

## 5   Discussion: trackers complementary

Now that we better understand what each tracker entails, we can discuss the benefits of using them in isolation or simultaneously. In particular, we keep in mind our objective to apply these trackers to simulation results for which we cannot make a pointwise comparison with observations.

It would be tempting to aggregate the four trackers catalogs. For example, using the union of all trackers would maximize the POD up to 92 %. However, it would also increase the FAR to 42 %. The opposite approach to use the intersection of all trackers would reduce the FAR down to 6 %, but also cut the POD down to about 65 %. Obviously, none of these simple approaches is ideal by itself. Likewise, taking the mean value of the metrics might seem attractive: for example, in our case, the mean



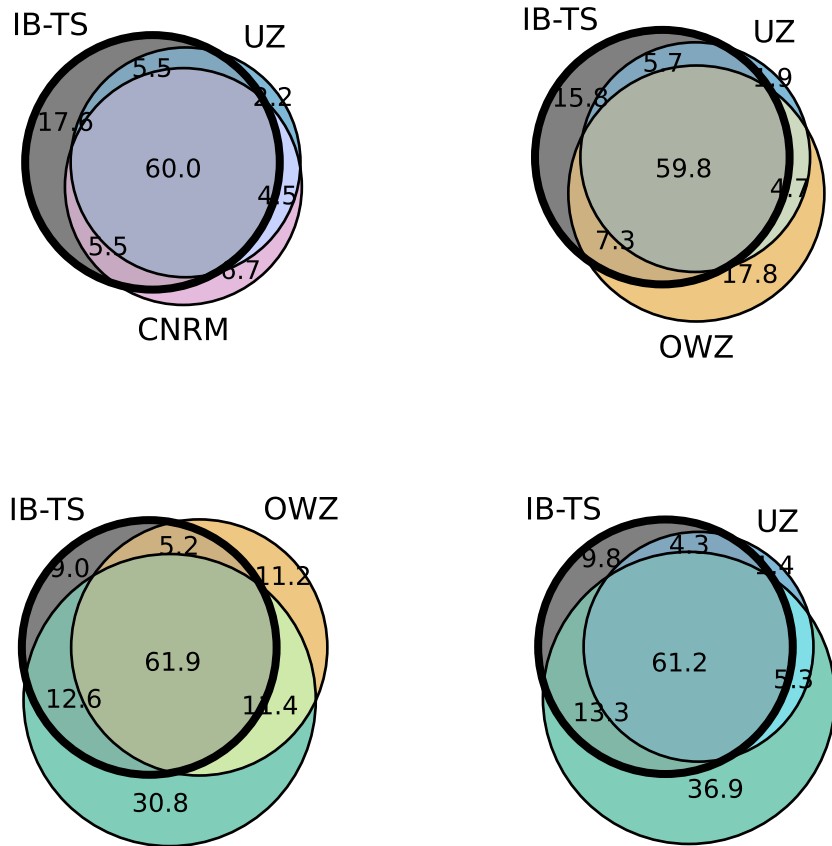

**Figure 12.** Venn diagrams representing common tracks between each algorithm and IB–TS. The figures in the circles are the annual frequency of each group.

value of the storms frequencies features an almost vanishing bias (see table 3). But as shown by the large associated scatter,
this is not significant and only the result of our specific choice of trackers. This approach, though, helps identify aspects of the
detected trajectories that are robust (i.e., tracker independent). As shown above, this is the case for the negative ACE biases,
which result from intrinsic difficulties for TCs to amplify enough in models and reanalysis.

An alternative might be to choose the "best" tracker based on its ability to minimize a given metric or a set of metrics. For
example, OWZ minimizes the bias on frequency and TC days (table 3). This is because, for OWZ, the number of missing TCs
is almost equal to the number of FAs. In addition, FAs and missing storms have similar global properties in terms of intensity,
latitude of pressure maximum, seasonal cycle, and track durations (compare the red and green distributions in figure 1). It
means that, on the global scale, FAs can be thought of as a substitute for the missing. This property of OWZ was already noted



in ERAI by Bell et al. (2018) and appears to hold here for the particular case of ERA5. However, we caution that this nice global agreement hides a significant regional variability: For OWZ, the number of missing TCs largely outnumbers the number
of FAs in the Eastern North Pacific, a bias that is compensated by the larger number of FAs in both the Southern Indian and Pacific oceans (see figure 5). These differences point to regional biases, and it is difficult to anticipate how and whether these biases would translate in any numerical simulation.

We found that an interesting method to build a quick and robust intuition about the similarities and differences between the different trackers is to use Venn diagrams, such as shown in figure 12. Venn diagrams plot as many circles as ensembles
considered. Their respective area is proportional to the number of detected tracks. The different circles superpose according to the detections they have in common. They immediately identify the common detections between two algorithms and their respective FAs and misses. In figure 12, we compiled four such diagrams spanning the different options to combine different pairs of trackers. We take IB–TS as the reference for all cases, although we note that other choices can be made.

The most obvious result is that there is a large pool of observed TCs that all trackers detect: there are 3510 tracks in total in
IB–TS and about 2400 of them are detected by all trackers, regardless of the Venn diagram considered. This result is simply graphically reflecting the large PODs we found for all trackers.

Second, figure 12 (top left panel) shows that UZ and CNRM are nearly identical: the number of detections they have in common vastly outnumbers the number of tracks detected by one tracker without being detected by the other. This overlap explains the similar properties noted above for these trackers, both for the globally integrated metrics and properties and for
the geographical distributions of detected TCs. We conclude that UZ and CNRM are essentially identical as far as tracking TCs is concerned. Venn diagrams nicely illustrate the increasing sensitivity of the different trackers: OWZ is more sensitive than UZ and the CNRM tracker (top right panel) but is itself less sensitive than TRACK (bottom left panel). Both diagrams also highlight the increasing number of FAs with sensitivity already discussed above.

One approach to exploit the respective strengths of the trackers is to combine two of them. The third Venn diagram (bottom
left panel) illustrates such a possibility: the idea is to combine the low FAR of UZ with TRACK's ability to follow TCs extended lifecycles. Combining UZ and TRACK hits only reduces the POD to 70% but retains UZ's low FAR. By removing TRACK FAs, which are frequent close to the equator, this approach could give stronger support to an analysis of the link between North Atlantic TCs and African Easterly Waves such as illustrated of figure 9 and still benefit from TRACK ability to probe TCs early trajectories before they are amplified enough to be detected by UZ.

We end by noting that if one is only interested in the strongest tracks in a simulation, the detection skills of all trackers are identical and nearly perfect. As already described above, no detected track beyond category 2 (included) is a false alarm, and TC observed with a category 4-5 are found whatever the tracker. In this case, other properties might become more important, such as the ability to track a larger part of the lifecycle provided by OWZ and TRACK.



## 6 Conclusions

In the present paper we have applied four tracking algorithms to ERA5 over the period 1980-2019. These trackers are UZ (Zarzycki and Ullrich, 2017; Ullrich et al., 2021), OWZ (Tory et al., 2013b; Bell et al., 2018), TRACK (Hodges, 1994) and the CNRM tracker (Chauvin et al., 2006). The Probability of Detections evaluated against IBTrACS range from 75 to 85%, and the False Alarms Rates vary between 19 up to 60%. Tracks missed by the trackers correspond mostly to weak tropical storms. We did not investigate their properties in details, but one possibility is that missing tracks corresponds to storms that were not

reanalyzed with sufficient intensity in ERA5. False alarms correspond to either weak tropical disturbances or extra-tropical cyclones. We derived two objective filtering methods to target these extra-tropical false alarms. The first one is based on the environment of the tracks: It relies on the relative positions of the detected tracks and the upper troposphere subtropical jet (STJ). The second one is based on the third Hart phase space parameter, the upper-level thermal wind (VTU): It allows to determine whether the core of the storm is warm or cold. Both post-treatments can be applied identically to any catalogue of

TC tracks. For the four trackers we used for this study, we found a dramatic reduction of the false alarms for all trackers except OWZ: False alarm rates range from 9 to 36% after post-treatment, which correspond to reductions up to 76%.

We then studied how several traditional metrics biases depend on the choice of the tracker. TC frequencies are highly sensitive to the algorithm. This is consistent with the study of Horn et al. (2014) who found that the intensity threshold drives the difference of TC frequencies found using several physics-based trackers. Our analysis shows that this is true when including

dynamics-based trackers as well. The number of TC days also varies with the algorithm: tracks mean duration is smaller than the observation for UZ and CNRM, similar for OWZ, and longer for TRACK. Both metrics sensitivity reflects the large variability in trackers' selectivity regarding weak storms. They are also consistent with Raavi and Walsh (2020), who found that CSIRO features simultaneously lower TC frequencies and shorter tracks than OWZ. Other metrics do not suffer from that variability, though. The ACE is almost uniform across trackers because it is mostly sensitive to the strongest TCs, for which all

trackers agree.

It should be noted that these global scores are heavily weighted by the most active oceanic basin, namely the western north Pacific ocean. TC frequencies in that particular basin compare well with the observations and the scatter across trackers remains moderate (Figure 6). This is more an exception rather than the rule. In the south Indian and south Pacific oceans, TRACK bias in TC frequency is positive and large, while the other trackers are close to IB-TS. In the eastern north Pacific, TRACK bias in

TC frequency is small, while the other trackers biases are negative and large. The North Atlantic ocean is peculiar because all trackers feature negative biases. Contrasted results are also found for the number of TC days. They are not easy to understand. They may be due to differences in reporting methods by each agency and/or to the varying resolution of the observations available for assimilation. TCs may also have different intrinsic properties in the different oceanic basins. It will be interesting to investigate whether these geographical differences hold in model results in order to disentangle between these alternatives.

Finally, it is important to keep in mind that deriving detection scores implicitly implies that there exists some sort of binarity between tropical disturbances, tropical storms and tropical cyclones. Of course, this is not the case in reality where these meteorological systems form a continuum. To some extent, trackers' thresholds are arbitrary and artificially create a strict limit



in the grey zone that separates tropical disturbances, storms and cyclones. One should not forget that these limits are trackers design choices that reflect the goals of their designers. OWZ was created to study precursors and to be resolution-independent
(Tory et al., 2013c). TRACK aims at detecting all vorticity perturbations, while TC identification is secondary (Hodges, 1994). UZ and the CNRM trackers were calibrated using a series of observed metrics on the reanalyses (Zarzycki and Ullrich, 2017). These design choices are reflected in past and present results and will affect future analysis of both reanalysis and models.

*Code and data availability.*  ERA5 data is available on the Copernicus Climate Change Service Climate Data Store (CDS, https://cds.climate.copernicus.eu/cdsapp#!/dataset/reanalysis-era5-pressure-levels). The IBTrACS database is provided by NOAA at https://www.ncdc.noaa.
gov/ibtracs/.
All the scripts used to produce the present paper's results are available at https://doi.org/10.5281/zenodo.6424432. These include the code to run the UZ and OWZ trackers and the original TRACK and CNRM databases, the code for the post-treatment and the tracks matching python scripts for the whole analysis, the code to reproduce the figures, and finally, a copy of the v.0.5 of the dynamicoPy package used in the python scripts (https://github.com/stella-bourdin/dynamicoPy).

*



## Appendix A:  List of acronyms

| Acronym | Definition |
|---------|------------|
| TC | Tropical Cyclone |
| TS | Tropical Storm |
| MSLP | Mean Sea-Level Pressure |
| SSHS | Saffir-Simpson Hurricane Scale |
| NH | Northern Hemisphere |
| SH | Southern Hemisphere |
| IBTrACS | International Best Track Archive for Climate Stewardship |
| IB–TS | Tropical Storm subset of IBTrACS |
| IB-TC | Tropical Cyclone subset of IBTrACS |
| ERA5 | Fifth European ReAnalysis |
| UZ | Ullrich & Zarzycki |
| OWZ | Obuko-Weiss-Zeta |
| CNRM | Centre National de Recherches Météorologiques |
| FA | False Alarm |
| FAR | False Alarm Rate |
| POD | Probability of Detection |
| STJ | Sub-Tropical Jet |
| VTU | Upper-level Thermal Wind |
| ET | Extra-Tropical |
| ETC | Extra-Tropical Cyclone |
| NAtl | North Atlantic |
| WNP | Western North Pacific |
| ENP | Eastern North Pacific |
| SP | South Pacific |
| NI | North Indian |
| SI | South Indian |
| ACE | Accumulated Cyclonic Energy |

**Table A1.** List of Acronyms used in this article





## Appendix B: Supplementary figures and tables

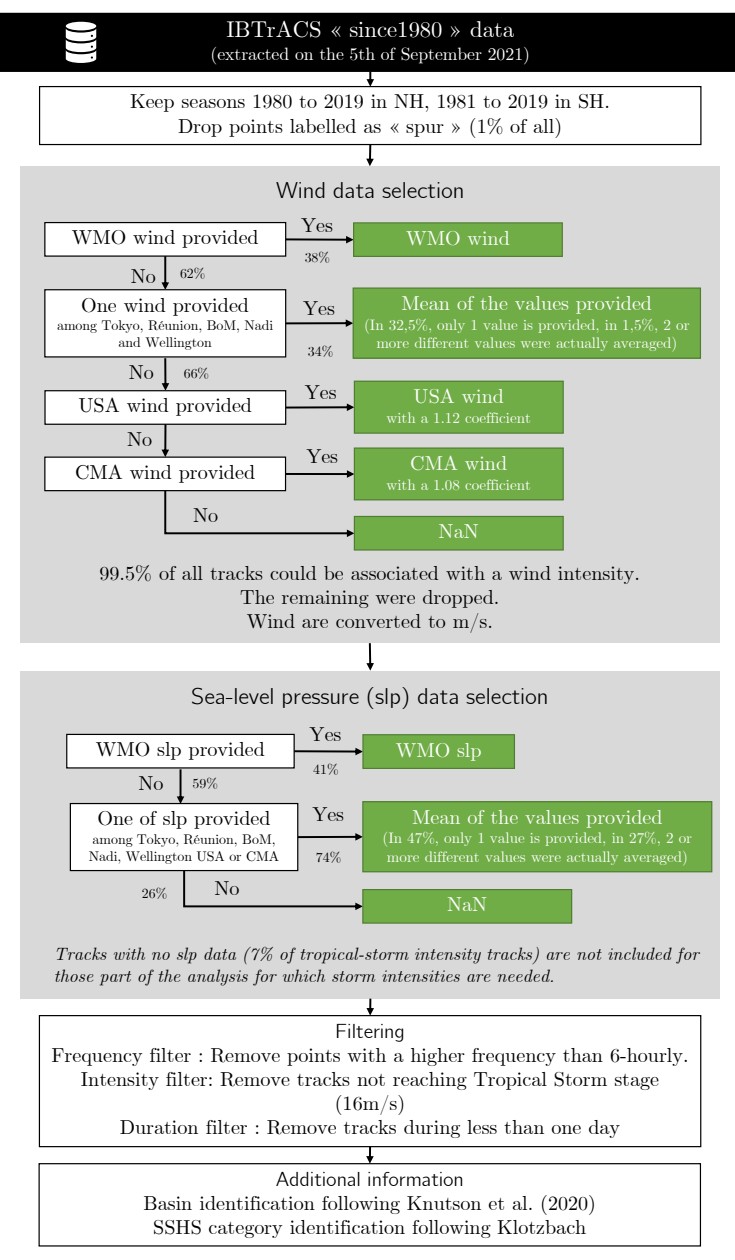

**Figure B1.** Workflow chart describing the treatment of the IBTrACS database in out study. The 1.08 coefficient to convert 3–minutes sustained winds to 10–minutes sustained winds was obtained using a linear regression on the data for which we had both.



| | | DetectNodes (or equivalent) | |
|---|---|---|---|
| | Local extremum | Candidates criteria | Merge distance |
| UZ | SLP minimum | SLP closed contour $2\,\mathrm{hPa}$ in 5.5° GCD $Z_{300-500}$ closed contour $-58.8\,\mathrm{m^2\,s^{-2}}$ in 6.5° GCD | 6° |
| OWZ | $OWZ_{850}\,maximum$ | $OWZ_{850} \geq 5\times 10^{-5}\,\mathrm{s^{-1}}$ $OWZ_{500} \geq 4\times 10^{-5}\,\mathrm{s^{-1}}$ $r_{950} \geq 70\%$ $r_{700} \geq 50\%$ $q_{950} \geq 10\,\mathrm{g\,kg^{-1}}$ $\mathrm{vws} \leq 25\,\mathrm{m\,s^{-1}}$ | 5° |
| TRACK | $\zeta_{850}$ maximum | $\zeta_{850} \geq 5\times 10^{-6}\,\mathrm{s^{-1}}$ | — |
| CNRM | SLP minimum | $\zeta_{850} \geq 1.5\times 10^{-4}\,\mathrm{s^{-1}}$ $u_{850} \geq 5\,\mathrm{m\,s^{-1}}$ $\sum_{700,500,300}\bar{T} \leq 1\,\mathrm{K}$ $\bar{T}_{850} - \bar{T}_{300} \leq 1\,\mathrm{K}$ $u_{300} - u_{850} \leq 5\,\mathrm{m\,s^{-1}}$ | 10 grid-points |

| | | | StitchNodes (or equivalent) | | | Relaxation |
|---|---|---|---|---|---|---|
| | Maximum distance | Maximum gap | Minimum duration | Additional criteria | Criteria duration | |
| UZ | 8°GCD | 24h | 10 time steps (54h) | $u_{10} \geq 10\,\mathrm{m\,s^{-1}}$ $|\phi| \leq 50°$ $z \leq 150\,\mathrm{m}$ | 54h | — |
| OWZ | 5° GCD | 24h | 9 time-steps (48h) | $OWZ_{850} \geq 6\times 10^{-5}\,\mathrm{s^{-1}}$ $OWZ_{500} \geq 5\times 10^{-5}\,\mathrm{s^{-1}}$ $r_{950} \geq 85\%$ $r_{700} \geq 70\%$ $q_{950} \geq 14\,\mathrm{g\,kg^{-1}}$ $\mathrm{vws} \leq 12.5\,\mathrm{m\,s^{-1}}$ $u_{10} \geq 16\,\mathrm{m\,s^{-1}}$ | 9 time-steps (48h) 1 time step | — |
| TRACK | — | None | 8 time steps (2 days) | $\zeta_{850} \geq 6\times 10^{-5}\,\mathrm{s^{-1}}$ $\zeta_{850} - \zeta_{250} \geq 6\times 10^{-5}\,\mathrm{s^{-1}}$ $\zeta_{850,700,600,500,250} \geq 0\,\mathrm{s^{-1}}$ $|\phi_{first}| \leq 30°$ | 4 time-steps (1d) | — |
| CNRM | — | None | 4 time steps (1 day) | — | — | With $\zeta \geq 200 \times 10^{-6}\,\mathrm{s^{-1}}$ |

**Table B1.** Synthesis of the trackers' criteria. All subscript but 10 correspond to pressure levels in $\mathrm{hPa}$





**Figure B2.** Characterisation of the false alarms of each algorithm. First line of distributions correspond to the minimum sea-level pressure, second line to the latitude of the pressure minimum, and third line to the track duration. In all plots, blue distribution correspond to the hits, orange to the false alarms before the post-treatment, and red to the false alarms remaining after the post-treatment.



## Appendix C: TempestExtremes code and OWZ adaptation

### C1  UZ

The code for UZ is exactly the same as in Ullrich et al. (2021), we only adapted to our own data infrastructure.

**Listing 1.** DetectNodes code for UZ

```
    ~/tempestextremes/bin/DetectNodes \
        --in_data_list flist_$yr$mth.tmp \
        --out $NODES_FILE \
        --timefilter "6hr" \
        --searchbymin msl \
        --closedcontourcmd "msl,200.0,5.5,0;\
_DIFF(geopt(300millibars),geopt(500millibars)),
-58.8,6.5,1.0" \
        --mergedist 6.0 \
        --outputcmd  "msl,min,0;\
_DIV(geopt(1000millibars),9.81),max,0;\
_VECMAG(u10,v10),max,2" \
        --latname latitude --lonname longitude
```

**Listing 2.** StitchNodes code for UZ

```
~/tempestextremes/bin/StitchNodes \
        --in_list nodeslist.tmp \
        --out tracks/ERA5_UZ.csv \
        --in_fmt "lon,lat,slp,zs,wind10" \
        --range 8.0 \
        --mintime "54h" \
        --maxgap "24h" \
        --threshold "wind10,>=,10.0,10;
lat,<=,50.0,10;lat,>=,-50.0,10;
zs,<=,150.0,10" \
        --out_file_format "csv"
```

### C2  OWZ

For this study, we adapted the OWZ algorithm presented in Section 2.3 to be used in the TempestExtremes (TE) framework (Ullrich and Zarzycki, 2017; Ullrich et al., 2021). Doing so involved a change in part of the methodology, and the arbitrary choice of some criteria that are required by the TE framework.

#### C2.1  Original algorithm

635 In Bell et al. (2018), the OWZ algorithm applied on ERA-Iterim data is described as follows:




   (i) "Each grid point at each time-step is assessed based on the initial thresold values;

  (ii) Clusters (or "clumps" in Tory et al. 2013) are formed by gathering neighbouring points that satisfying the initial thresholds, that are supposed to represent a single circulation ate that point in time;

 (iii) Circulations from step (ii) are linked through time by estimating their position in relation to the circulation's expected position based on an averaged 4°x4° steering wind at 700hPa;

 (iv) Tracks are terminated when no circulation match is found in the next 2 time-steps within a latitude-dependent radius ( 350km);

  (v) Tracks are declared TC if the core thresholds are satisfied for five consecutive 12h periods."

The thresholds are provided in their Table 1.

Tory et al. (2013b) further specifies that "clumps in close proximity are reduced to one clump by discarding the weaker or smaller clumps", and that each clump must satisfy a set of clump conditions: "a minimum size limit, and a land-impact condition". the minimum size limit is two gridpoints, the radius to look for weaker or smaller clumps is 550km. The land-impact condition tests whether the point is over the land or the ocean. In this paper, the latitude-dependent radius of step (iv) "varies linearly from 600 to 400 km between 15° and 30° latitude in both hemispheres, with constant values outside this latitude band", which does not match with the 350km specified by Bell et al. 2018.

### C2.2 TempestExtremes Adaptation

The TempestExtremes nodal feature detection framework (DetectNodes + StitchNodes) does not work with the same paradigm, but still allows us to implement a very similar algorithm. The fundamental principle in DetectNodes is to track a local extremum of one variable. It can then merge all extrema in a given radius ($r_{merge}$) into the largest one, and the position of the extremum is considered the center of the detected feature. One can also add discriminatory criteria, either (i) in terms of thresholds to be satisfied for given variables at the gridpoint of the extremum, or by one gridpoint in a given radius $r_{threshold}$ from the center, or (ii) in terms of a closed-contour. (see Ullrich et Zarzycki 2017). The subsequent StitchNodes uses a nearest-neighbors approach to link consecutive points within a maximum distance $r_{range}$. In this command, one can also allow for a gap to exist in the track, and check additional thresholds that must be satisfied for a given number of points in order to validate the track.

Here we choose to look for a local maximum of OWZ, and to merge all weaker maxima in a 5° great-circle-distance ($\approx$ 550km in the original algorithm). Thresholds from the original algorithm must be satisfied by at least one gridpoint in a $r_{thresholds}$ radius (that will be determined from the following sensitivity tests). In the StitchNodes command, we look for consecutive points within a $r_{range}$ radius (that will also be determined from the following sensitivity tests). A 24h gap is allowed, corresponding to the "next 12h time steps"). On top of this, core thresholds are assessed within the same $r_{thresholds}$ radius, and must be satisfied for at least nine 6-hourly time-steps. Core thresholds include the "land-impact condition", that we implement using the land-sea mask, and considering as ocean points with less than 50% of land. The minimum duration is set to 48h, corresponding to the nine 6-hourly time-steps, so that it is not a discriminatory criterion but helps accelerate the computation.



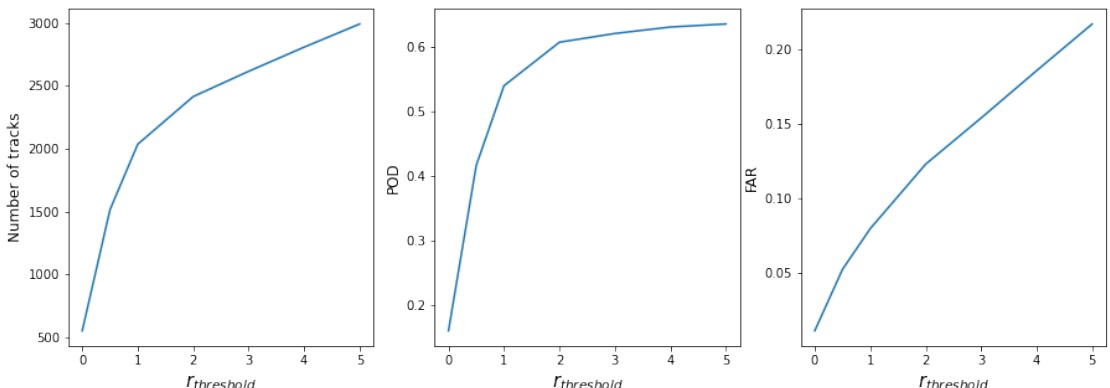

**Figure C1.** Sensitivity of the number of tracks, the POD, and the FAR of OWZ for different values of $r_{threshold}$.

At this stage, we have 2 parameters left to determine : $r_{thresholds}$ and $r_{range}$, for which we conduct independent sensitivity
670 tests.

### C2.3 Sensitivity analysis

In the original algorithm, the thresholds must all be satisfied in the same gridpoint. However, it was used on ERA-Interim data interpolated onto a $1° \times 1°$ grid, whereas ERA5 data presents itself with $0.25° \times 0.25°$ grid points. One can expect this higher resolution might allow for the formation of an eye in the circulation, so that all the thresholds might be verified in the wall
675 rather than in the center of the circulation. We test 7 values for $r_{thresholds}$ : $0°$ (thresholds must be passed at the center), $0.5°$, $1°$.

The 350km range in Bell et al. 2018 corresponds to $3°$, whereas the [400; 600km] range in Tory et al. 2013 correspond to $[3.5°;5.5°]$. In UZ, the range is set to $8°$. We test $r_{range}$ values between 3 and $8°$.

To assess the sensitivity of the detection to these thresholds, we compute the number of tracks detected. We also pair detected
680 tracks with observed IB-TS tracks following the procedure described in 2.4, and compute the false alarm rate and the probability of detection.

The sensitivity to the $r_{range}$ parameter is low (not shown), in accordance to results on UZ by Zarzycki and Ullrich (2017). We choose $r_{range} = 5°$ in the middle of Tory et al. 2013 range. Figure C1 shows the sensitivity of the three metrics to the $r_{thresholds}$ parameter. We can see that for $r_{threshold} \geq 2$, the POD saturates, and all additional tracks are false positive. For
685 this reason, we keep $r_{threshold} = 2$.

### C2.4 Code

**Listing 3.** DetectNodes code for OWZ

```
~/tempestextremes/bin/DetectNodes \
```



```
          --in_data_list tmp/flist_${date}.txt \
          --out $NODES_FILE \
          --timefilter "6hr" \
          --searchbymax "owz(850millibars)" \
          --mergedist 5.0 \
          --latname latitude --lonname longitude \
          --thresholdcmd "owz(850millibars),>=,0.00005,2;\
owz(500millibars),>=,0.00004,2;\
r(950millibars),>=,70,2;\
r(700millibars),>=,50,2;\
_VECMAG(_DIFF(u(850millibars),u(200millibars)),
_DIFF(v(850millibars),v(200millibars))),<=,25,2;\
q(950millibars),>=,0.01,2" \
          --outputcmd "owz(850millibars),max,2;\
owz(500millibars),max,2;\
r(950millibars),max,2;\
r(700millibars),max,2;\
_VECMAG(_DIFF(u(850millibars),u(200millibars)),
_DIFF(v(850millibars),v(200millibars))),min,1;\
q(950millibars),max,2;\
msl,min,3;\
_VECMAG(u10,v10),max,2;\
_VECMAG(u(925millibars),v(925millibars)),max,2;\
vo(850millibars),max,2" \
          --searchbythreshold ">0"
```

**Listing 4.** StitchNodes code for OWZ

```
~/tempestextremes/bin/StitchNodes \
          --in_list tmp/nodeslist.txt \
          --out $file_name \
          --in_fmt
"lon,lat,owz850,owz500,r950,r700,vws,q950,slp,wind10,wind925,vo850" \
          --range 5.0 \
          --mintime "48h" \
          --maxgap "24h" \
          --threshold "owz850,>=,0.00006,9;\
owz500,>=,0.00005,9;\
r950,>=,85,9;\
r700,>=,70,9;\
vws,<=,12.5,9;\
q950,>=,0.014,9;\
wind10,>=,16,1" \
          --out_file_format "csv"
```




## Appendix D:  Match characteristics

Here we validate our matching method through a few sanity checks. They show that the pairing methodology is not very sensitive.

Figure D1 shows the distribution of the numbers of overlapping time-steps, i.e. the time for which two paired tracks are closer than 300km. By construction of our method, there must be at least 1 of them. It is compared to the distribution of lifetime in IB-TS. Figure D2 shows the proportion of the observed lifetime matching the corresponding ERA5 track, when it

exists. Figure D3 show the distribution of the distance between the observed and detected tracks, averaged over the overlapping timesteps for each pair of tracks. Each distribution is provided for each tracker as a boxplot that indicates the 1st, 25th, 50th, 75th and 99th percentiles.

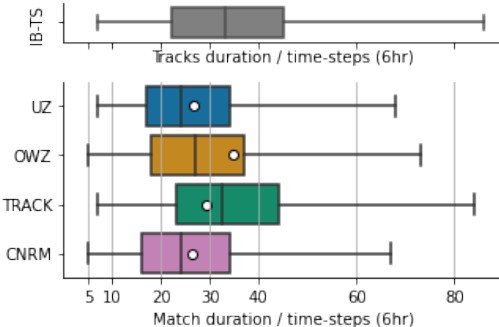

**Figure D1.** Distribution of the duration of the overlap between matching detected and observed tracks. Whiskers display the 1st and 99th percentiles, and white points show the mean of the distributions. Outliers are not shown.

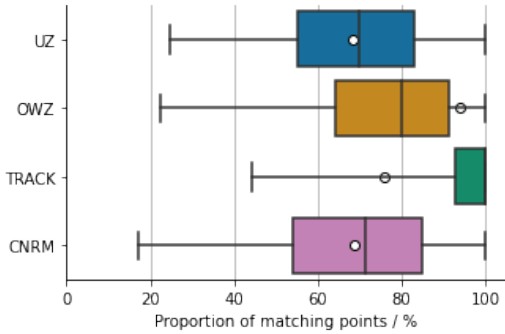

**Figure D2.** Distribution of the duration of the overlap between matching detected and observed tracks. Whiskers display the 1st and 99th percentiles, and white points show the mean of the distributions. Outliers are not shown.

Figure D1 show that despite the fact that our methodology imposes only 1 overlapping points, more than 99% of the pairs, whatever the tracker, match for at least 5 time-steps. It shows that the matching is not sensitive to this threshold. In fact,





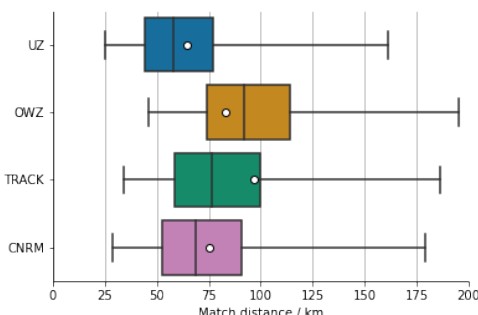

**Figure D3.** Distribution of distance between matching detected and observed tracks. Whiskers display the 1st and 99th percentiles, and white points show the mean of the distributions. Outliers are not shown.

figure D3 shows that in most cases observed tracks are matched for more than half of the observed lifetime (figure D3). This proportion relates to the mean detected lifetime by each tracker displayed in figure 7. Moreover, the matching distance is of the order of a few grid cells, inferior or close to $100\,\mathrm{km}$ whatever the trackers, with a slightly better accuracy of the trackers that use SLP as their center. All this gives us confidence in the fact that the tracks that are paired together are indeed corresponding, because they are close to one another for a significant part of their lifetime.

Figure D3 (TRACK line) can be compared to Hodges et al. (2017), who found that the TRACK match distance was about $1°$ for other reanalyses tracked with TRACK. The improvement can be related to the increase in resolution.



*Author contributions.* SB designed and carried out the study under the supervision of SF. All authors provided critical feedback and helped shape the whole study. WD ran the CNRM tracker under the supervision of JC and FC. SB and SF prepared the manuscript, with input from all co-authors. SB did the figures. SF obtained the fundings.

*Competing interests.* The author declare no competing interests.

*Acknowledgements.* The research is supported by public funding to the CEA and CNRM. S. Bourdin and S. Fromang were also financially supported by the EUR IPSL-Climate Graduate School through the ICOCYCLONES project. J. Cattiaux, F. Chauvin and W. Dulac were also financially supported by the CNRS/INSU trough the LEFE/CYPRESSA project.

The authors are grateful to K. Hodges for providing the TRACK dataset, for insightful scholar discussion and literature provision and for

his review of an earlier version of this manuscript. The authors thank to P. Ullrich and C. Zarzycki for their help regarding TempestExtremes.

S. Bourdin thanks colleagues at the LSCE for helpful discussion and feedback, and especially R. Noyelle for the idea of using the Hart phase space.





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
