# Peer review of "Intercomparison of Four Tropical Cyclones Detection Algorithms on ERA5"

_EGUsphere, 2022_

## Referee Comment (RC1)

**Overall comments:**

In the current manuscript, the authors have compared the performance of four entirely different tracking scheme using high-resolution ERA5 reanalysis data compared to the observations (IBTrACS). In addition, they have implemented a common criterion in all the tracking schemes to remove extra-tropical storms in the detected tropical cyclone detections to reduce the False Alarm rates. The paper is very interesting and well written by accurately identifying the gap in the current literature that model projections on TC characteristics are sensitive to the underlying tracking scheme. This novel study on identifying pros and cons of different tracking schemes can help us to accurately choose one better tracking scheme or combination of tracking schemes that can help to improve the certainty of future climate model projections on TC characteristics. Although, the paper does not directly involve modelling, it gives a new method/idea in analyzing the future modelling results related to extreme weather events (Tropical cyclones) that is important for global numerical modelling community, policy makers and risk assessment companies.

All the methods and assumptions are clear and clearly outlined in the manuscript. The results are sufficient to support the interpretations and conclusions. In addition, they can also check whether different tracking schemes can capture the interannual changes in the TC frequency due to ENSO. The study also sufficiently compared/contrasted with earlier studies and clearly indicated their novel contribution to the paper. Overall, the presentation of the manuscript and supplementary sections are clear with sufficient information about the codes to reproduce the results.

**Detailed comments:**

Abstract is missing the research problem of the article. Please include the goal of your research work in the abstract.

Paragraph 15 is not clear. Needs to explain the conclusion here more clearly. Does the author mean that we need to select one or a combination of few trackers with better performance and average the result.?

Also, in the paragraph 75 the word "we used" has been used frequently, please reduce the usage of that. Also, in the entire draft we find the word "we used" is repeated. Instead use "we employed", "we utilized".

At paragraph 40, you have mentioned that physics-based trackers embed a wind threshold. Please be precise here which wind threshold do you mean? I guess it should be 10m winds.

At the paragraph 45, mention that OWZ **tracker** was found to produce better results across a wide range of resolutions instead of just OWZ.

At line 70, Dulac et al., ? year is not mentioned here.

In the paragraph 130, the following sentence is not clear: Nevertheless, it has recently been assessed as having similar performances for a range of metrics (Zarzycki et al., 2021; Roberts et al., 2020a).

Coming to the methods section on TC trackers: It would be good to provide a table describing different tracker input variables, main idea of the design of the tracking scheme, spatial and temporal resolution requirements of the tracker variables, etc.

In the description of the post treatment methods, where you have used two different methods and only focus mainly on the STJ method in the entire manuscript. So, I suggest maybe you can keep the detailed description explaining the VTU method in the supplementary section.

In the Discussion section, you have introduced Venn diagrams concept. I suggest you to give some details about how to build the Venn diagrams in the methods section.

---

## Community Comment (CC1)

Referee review of Bourdin et al: "Intercomparison of four tropical cyclone detection algorithms on ERA5"

**Overall comments**

In this article, the authors describe a comparison of tropical cyclones in ERA5, derived using four different tracking methods, against observations. The strengths and weaknesses of the different tracking methods are evaluated which tracks are found or not found compared to observations, and what properties these different tracks have. The authors also devise some post-processing to remove some of the false alarms.

This is a really valuable contribution to the literature on tropical cyclone analysis, since many papers on tropical cyclones use different methods and a thorough comparison of some of these methods is overdue. This is also important given the uncertainty in future projections of tropical cyclones, and having better understanding of strengths and weaknesses of trackers is useful for understanding some of these uncertainties. The paper is very well written, logically organised, and contains inciteful analysis techniques to better understand the tracking methods.

I have a several broader questions that I'd like to see clarified in the text.

For the benefit of potential users of the tracking algorithms I think it would be useful to have some mention of their respective complexities, for example ease of obtaining and use. Having used several of these trackers myself, there is a very wide range of cost and complexity involved (some can simply be downloaded and run, others not), and I think that this information together with the scientific quality of the tracker outputs would be valuable to communicate.

You choose to use six hourly data from ERA5 for the tracking. I'd like to see this justified more robustly (that is, I am not expecting you to retrack using hourly data). I could have imagined that this would have been an excellent opportunity to use the available hourly data from ERA5, hence removing uncertainties that come from using six hourly data (particularly from stitching points of the tracks together, and from missing points within a track), and helping to clarify whether it remains good enough for climate models only to produce six hourly data for tracking, or whether key aspects of the tropical cyclones are missed. Hourly tracks could be just as easily compared to six hourly observations after all.

The North Atlantic seems to be a difficult basin for all the trackers as they all have a low bias. Is there a reason for this that you can suggest? Since many climate models also struggle with TCs over the Atlantic I feel this may be an interesting observation.

**Detailed comments**

L20: bottleneck impediment – maybe choose one of these words, rather than both

L69: "in a forthcoming paper…" – perhaps you could just say "in future work", if the manuscript is not available yet.

L128: As in the overall comments, could you add a sentence to justify using six hourly data? Since there is hourly reanalysis data, is this primarily because IBTrACS generally does not have hourly data (although you could match the tracks even if the times are not the same)? Do you think that the tracks would be more accurate using hourly data, certainly the uncertainty from gaps and distances

would be smaller? We found in recent work on Mediterranean cyclones that using hourly data is important, and it would be interesting to hear something about this, given that (by historic default?) most models produce six hourly data.

L202: Dulac et al – again, should you cite if not available?

Figure 1: Can you clarify what the y-axis is? Is it normalised frequency, or percentage?

Figure 3: could you clarify the caption, it is not quite clear.

L306: I think you should have a citation for the statement "…potential poleward shift with climate change".

Figure 8: Maybe clarify – does offset delay mean when the storm finishes (lysis)?

Figure 9: What does OWZ look like in terms of precursors, since you don't show it but I think it is one of its selling points? Given it is mslp-based, UZ cannot really detect over land, but OWZ might be able to, or is TRACK the only one? Perhaps you could add a figure to any supplementary data, I think this would help to distinguish the utility of the trackers or combinations of them.

L479: But Patricola et al. filtered out AEWs and found no change in TCs – how does this compare to your statement?

Figure 12: I really like the Venn diagram, but some of the numbers are difficult to read, could you make them a little clearer?

L511: Are there any characteristics of observed storms that make all four trackers miss (92% POD means 8% miss)?

L544: "The third Venn diagram" – I think you are referring to the fourth one, bottom right.

L550: This paragraph is true for the reanalysis, does it necessarily hold for climate models? Could models' representation of TCs be distorted (since there is no data assimilation) such that different trackers would not necessarily find the stronger TCs, or not?

L581: "Contrasted results …varying resolution of observations" – it is not clear to me what you are trying to say in these several sentences. Perhaps you could rewrite to make clearer? I really don't know what varying resolution of observations means, do you mean how frequent they have been over time, or how densely observed pre-/post- satellite era?

L585: binarity – my dictionary does not have this word. Perhaps binary choice?

L587: "To some extent, trackers' thresholds are arbitrary" – this may be true, but could one also argue that observational classification of TCs is also somewhat arbitrary, satellite images with an interpreted core of the storm or other methods? Perhaps you disagree, but maybe you could qualify the statement a little.

L887: You could mention here that understanding precursors is becoming a key question for future projections (in particular to understand diverging projections of increase or decrease in frequency in future), so having trackers able to identify such features is really important.

Table B1: For TRACK, I think here you are using the version averaging vorticity across 850,700,600 hPa, hence rather than eta_850 I think it should be eta_bar_T63, as you have written on L192.

Figure B2: What is the difference between Fig. B2 and the earlier, similar one, can you clarify?

Looking at the algorithm parameters within StitchNodes, the different trackers have different minimum durations set. But duration (lifetime) of the tracks are compared in the results section. Would it not have made more sense to use the same minimum duration across all trackers?

**Technical corrections**

Pacific Ocean should be capitalised – in various places in the manuscript.

L272: "extrem" → "extreme"

Figure 6 caption: missing a ( for "top panel)".

L409: "..for each tracker.." or "..for each of the trackers.."

L447: "maximum 10 meter wind speed"

L559: "..missing tracks correspond to"

Figure B1: Second to bottom box: "Remove tracks lasting less than one day"

L638: "single circulation at that point"

L657: Ullrich et Zarzycki

L669: several times in this section you have "r_thresholds" when I think you mean r_threshold.

L677: Several references in this section around here have lost their formatting, e.g. Bell et al. 2018 rather than Bell et al. (2018).

L738: Figure D1 shows that…

---

## Author Response (AR1)

**Response to reviews**

About *Intercomparison of four tropical cyclone detection algorithms on ERA5*

Stella BOURDIN on behalf of the authors

July 2022

Below are reproduced the answers we made to both reviewers' comments.

**1 Malcolm J. Roberts**

**1.1 Major comments**
* * *
**Comment 1**

For the benefit of potential users of the tracking algorithms, I think it would be useful to have some mention of their respective complexities, for example ease of obtaining and use. Having used several of these trackers myself, there is a very wide range of cost and complexity involved (some can simply be downloaded and run, others not), and I think that this information together with the scientific quality of the tracker outputs would be valuable to communicate.
* * *
We had been wary of making subjective comments about the ease of use of the different trackers. However, your comment made us realize that we can do it in an objective quantitative manner, and we agree that this could be useful to potential TC trackers users, for which our paper was intended.
We suggest adding the following table to the discussion, along with the associated comment:

|  | UZ | OWZ | TRACK | CNRM |
|---|---|---|---|---|
| Implementation | TempestExtremes (Command-line) | Python + TempestExtremes | Parameters files and shell scripts interfacing FORTRAN | Shell scripts interfacing FORTRAN |
| Computation time | 30 minutes | 13 minutes + 20 minutes | 30 minutes* | 1h30 |
| Parallelization available | MPI | MPI | Embarassingly parallel over each year. | Embarassingly parallel over time. |
| Open Source | Yes | Yes | No | No |

Table 1: Comparison of the different tracker implementations. Computation times are orders of magnitude, of the time necessary to track TC in one hemisphere over one year, sequentially. They might vary with different machines and setups. (* Personal communication from K. Hodges)

Finally, table 1 provides practical considerations on the implementation of each tracker. UZ is implemented using the TempestExtremes command-line software, which is easily parallelizable using MPI, and fully open-source. OWZ necessitates two steps, the first is the computation of the OWZ variable, which is done in Python, and the second is the tracking itself, done here with TempestExtremes (and therefore parallelized using MPI as well). We also benchmarked UZ on 1-hourly data, proving that the computation time scales linearly with temporal resolution. The code for OWZ is provided along with this paper. TRACK is run using shell scripts that read input from text files and run the FORTRAN code. Because it performs spectral filtering, it needs to be run

globally, and because of the stitching step that is not independent, it is tricky to split in the middle of a TC season. Therefore, the embarrassingly parallel potential is limited to parallelizing over the years. TRACK code is not open source but available upon request. CNRM is also implemented in FORTRAN, interfaced with shell scripts. It does not have any parallelization implemented so far, but it can be used embarrassingly parallel over space or time. In terms of computation time, UZ, OWZ, and TRACK are roughly equivalent, while CNRM requires about twice as much time. UZ presents the best potential for parallel acceleration, followed by OWZ, although with a slightly reduced potential because the TempestExtremes part corresponds to two third of its processing.
* * *
**Comment 2**

You choose to use six hourly data from ERA5 for the tracking. I'd like to see this justified more robustly (that is, I am not expecting you to retrack using hourly data). I could have imagined that this would have been an excellent opportunity to use the available hourly data from ERA5, hence removing uncertainties that come from using six hourly data (particularly from stitching points of the tracks together, and from missing points within a track), and helping to clarify whether it remains good enough for climate models only to produce six hourly data for tracking, or whether key aspects of the tropical cyclones are missed. Hourly tracks could be just as easily compared to six hourly observations after all.
* * *
We used 6-hourly data because our final goal is to use the trackers on simulations, for which, as is customary, we only have 6-hourly data available. Therefore, we thought best to be as close as possible to that usage. However, we do realize that there is an interest in comparing the results obtained with 1-hourly data, and an advantage to be taken from this data being available in ERA5. We ran the UZ tracker over the 1-hourly data between 1994 and 2020. Among the new tracks, there are about the same number of hits and false alarms. However, all of them are weak and short tracks, which fits with what we concluded that intense TCs are robustly detected and sensitivity to tracking methodologies lies in the gray zone of tropical storms. Quantitatively, using 1-hourly data, the POD changes from 78% to a value that ranges between 80% and 82% depending on the parameter that controls the distance between two consecutive points. Likewise, the FAR changes from 20% to a value comprised between 17% and 24%. Both changes are modest. We therefore conclude that it does not seem critical to use 1-hourly data for TC detection, which makes sense regarding our conclusions about the shortcomings of the trackers. It is probably more critical for medicanes because they are very short-lived compared to TCs and can have a warm-core phase that last only few hours and still make a lot of damage because of the proximity of land.

We changed the methods to better reflect this :

> We used data from the fifth generation of ECMWF Reanalysis (ERA5, Hersbach et al. 2020). ERA5 provides hourly estimates of atmospheric variables on a grid with 0.25° horizontal resolution from 1979 to the present day. **For the purpose of this work, we only used 6-hourly data from** 1980 **to** 2019 **(as in IBTrACS). The choice of using 6-hourly data has been made in regard of our final objective being to use the trackers on simulations, for which, as is customary, we only have 6-hourly data available. However, we check by running part of the tracking on 1-hourly data that the difference it makes is not important.**

We also added a comment on the computation time required in the discussion of the implementation.
* * *
**Comment 3**

The North Atlantic seems to be a difficult basin for all the trackers as they all have a low bias. Is there a reason for this that you can suggest? Since many climate models also struggle with TCs over the Atlantic I feel this may be an interesting observation.
* * *
We cannot think of a plausible explanation as is. Coming up with a potential explanation would require more analysis and is probably beyond the scope of this paper. We would prefer to avoid speculating and leave that for future work.

**1.2 Minor comments**
* * *
**Comment 1**

L20: bottleneck impediment – maybe choose one of these words, rather than both
* * *
This was indeed a typo that we corrected.
* * *
**Comment 2**

L69: "in a forthcoming paper..." – perhaps you could just say "in future work", if the manuscript is not available yet.
* * *
We had planned that this other manuscript would be submitted by the time of the review of the present one, but it has been delayed, so we will, as you suggest, switch to "in future work" like you suggest. In case the other manuscript is submitted by the time of the acceptation of this one, we will switch back to the actual reference.
* * *
**Comment 3**

L128: As in the overall comments, could you add a sentence to justify using six hourly data? Sincethere is hourly reanalysis , is this primarily because IBTrACS generally does not have hourly data (although you could match the tracks even if the times are not the same)? Do you think that the tracks would be more accurate using hourly data, certainly the uncertainty from gaps and distances would be smaller? We found in recent work on Mediterranean cyclones that using hourly data is important, and it would be interesting to hear something about this, given that (by historic default?) most models produce six hourly d
* * *
See Major comment 1.
* * *
**Comment 4**

L202: Dulac et al – again, should you cite if not available?
* * *
Due to changes in the manuscript of Dulac et al., we removed this reference altogether.
* * *
**Comment 5**

Figure 1: Can you clarify what the y-axis is? Is it normalised frequency, or percentage?
* * *
These are histograms, where the y-axis is a count. It has not been normalized in any way, so that the area under each curve is proportional to the number of tracks in each ensemble.
We suggest to update the caption this way:

> Histograms representing the properties of the Hits, the Miss, and the False Alarms tracks for each tracking algorithm. From left to right, the columns correspond to UZ, OWZ, TRACK, and the CNRM tracker, respectively. [...] The blue and gray colors correspond to the Hits and the Misses, respectively, for all plots. Raw False Alarms are shown in orange while we plot in red the False Alarms that remain after the post-treatment (see section **??** for details). **The histograms display counts that have not been normalized. Hence, the area under each curve is proportional to the number of tracks in each ensemble.**

Comment 6

Figure 3: could you clarify the caption, it is not quite clear.

The new figure block is reproduced below (Fig. 1), hopefully the new caption is more helpful.

[Figure]

[Figure]

Figure 1: **Illustration of the STJ (left panel) and VTU (right panel) post–treatment procedures. The left panel shows a close-up map of the WNP. It displays two tracks detected by the UZ tracker that occurred simultaneously (represented using square and diamond symbols). Also shown are the $200\,\mathrm{hPa}$ horizontal wind speed (red shadings), the $15\,\mathrm{m\,s^{-1}}$ zonal wind contour (dark red line) and the STJ limit at that time as defined in section ?? (white dashed line). The right panel displays both tracks in the Hart phase space diagram, also defined in section ??.**
**The track represented using square symbols on both panels features more than one point equatorward of the STJ limit (left panel) and in the upper part of the Hart diagram (right panel). It is thus classified as a genuine TC according to both post–treatment methods. In fact, it corresponds to Typhoon Mac (1982) as found using the track matching procedure described in section ??. By contrast, the track represented with diamonds on both panels lies poleward of the STJ limit (left panel) and in the lower part of the Hart diagram (right panel). It is thus classified as an ETC according to both post–treatment methods. It was indeed classified as a False Alarm according to the track matching algorithm. Finally, note that the gray points correspond to points that lie poleward of the STJ limit and are therefore labeled as extra-tropical by the STJ method.**

Comment 7

L306: I think you should have a citation for the statement "...potential poleward shift with climate change"

We added a reference to the IPCC's AR6 Technical Summary RN4. More precisely, in the TS.2.3 "Upper Air Temperatures and Atmospheric Circulation" they state that "Several aspects of the atmospheric circulation have likely changed since the mid-20th century, and human influence has likely contributed to the observed poleward expansion of the Southern Hemisphere Hadley Cell and very likely contributed to the observed poleward shift of the Southern Hemisphere extratropical jet in summer. It is likely that the mid-latitude jet will shift poleward and strengthen, accompanied by a strengthening of the storm track in the Southern Hemisphere by 2100 under the high CO2 emissions scenarios."

Comment 8

Figure 8: Maybe clarify – does offset delay mean when the storm finishes (lysis)?

We are unsure whether we understand your comment well, but we will try to clarify our approach here : Onset and offset do correspond somehow to the genesis and lysis respectively, but we were wary of using the terms

genesis or lysis, because the points at which trackers (and observation) start and end the tracks do not correspond to the actual cyclogenesis or cyclolysis. So we preferred to speak about "set" in the sense that a track is set on and off at some points. But we understand introducing additional vocabulary also makes it confusing. We changed these to "first point delay" and "last point delay" in order to be more precise and yet neutral.
* * *
**Comment 9**

Figure 9: What does OWZ look like in terms of precursors, since you don't show it but I think it is one of its selling points? Given it is mslp-based, UZ cannot really detect over land, but OWZ might be able to, or is TRACK the only one? Perhaps you could add a figure to any supplementary data, I think this would help to distinguish the utility of the trackers or combinations of them.
* * *
We added the figure with all four trackers in the appendix, reproduced below (fig 2), and added comments about OWZ and CNRM in the body of the text:

> One can use the ability of TRACK – **and, to a lesser extent, OWZ** – to detect storm tracks early in their life cycle to study TCs genesis locations. [... (see comment 10)] In IBTrACS, the first reported point of TCs tracks tends to be located in the eastern central Atlantic ocean, in agreement with the tracks detected in ERA5 by UZ (Figure 9, top and middle panels). By contrast, TRACK genesis locations extend further east and well over the African continent (Figure 9, bottom panels), i.e., well into the region where African Easterly waves develop. These differences in genesis locations illustrate TRACK's ability to follow from very early on the vorticity perturbations that later transform into genuine TCs, and potentially to associate these early perturbations with known atmospheric phenomena. **To this regard, OWZ is a middle ground between TRACK and UZ (Figure 8, left panel), and is able to find some precursors over land (Figure B3). CNRM is very similar to UZ(Figure 8, left panel), except it catches some precursors over land, probably because of its specific relaxation step taking into account vorticity only.** This property of TRACK **and OWZ** opens the way for studying the correlation between North Atlantic TC genesis and African easterly waves in ERA5, in the spirit of studies such as done by Thorncroft Hodges (2001), Hopsch et al. (2007) and Duvel (2021). [...]
* * *
**Comment 10**

L479: But Patricola et al. filtered out AEWs and found no change in TCs – how does this compare to your statement?
* * *
Our statement was indeed too strong and not taking into account enough the recent literature on the topic. We updated the statement to :

> One can use the ability of TRACK to detect storm tracks early in their life cycle to study TCs genesis locations. For example, in the North Atlantic, **although the exact role of African Easterly Waves (AEW) is still debated (Patricola et al. 2018), there is a correlation between AEW and cyclogenesis (Landsea et al. 1993, Avila et al. 2000), so that it is interesting to be able to probe the early parts of TCs life cycles.**

[Figure]

Figure 2: First observed/detected points in IB-TS, and each tracker. left column shows the first points along with the corresponding density. Right column overlays IB-TS tracks first point with ERA5 tracks first point as detected tracks by each tracker.

**Comment 11**

Figure 12: I really like the Venn diagram, but some of the numbers are difficult to read, could you make them a little clearer?

We have improved the figure's lisibility by putting out numbers that do not fit in their areas and adding arrows, as displayed below in Figure 3

[Figure]

Figure 3: Venn diagrams representing common tracks between each algorithm and IB–TS. The figures in the circles are the annual frequency of each group.

**Comment 12**

L511: Are there any characteristics of observed storms that make all four trackers miss (92% POD means 8% miss)?

In figure 4 below, we display a number of diagnoses about the tracks that are not detected by any tracker. We identified in the paper that the tracks missed by all trackers correspond to weak and short tracks, and here we can see that the ones that they all miss correspond to the very short and very weak end of the spectrum. They are unequally distributed among basins, but this distribution is similar to the Hits/Miss ratio for each tracker in general : The union has lower PODs in the ENP (84%) and NATL (82%) as is observed independently for all trackers, whereas SI, NI and WNP has respective POD of 96%, 96% and 95% illustrative of the fact that every tracker scores well for these basins.

[Figure]

Figure 4: Characteristics of the 311 tracks missed by all trackers (orange) compared to those of the hits (blue)

We modified the following sentence accordingly to highlight that point:

> For example, using the union of all trackers would maximize the POD up to 92 %, **with the common 8 % of observed storms missed by all trackers corresponding to the weakest and shortest IB-TS storms**.

**Comment 13**

L544: "The third Venn diagram" – I think you are referring to the fourth one, bottom right.

Yes indeed, thank you for your vigilance.

**Comment 14**

L550: This paragraph is true for the reanalysis, does it necessarily hold for climate models? Could models' representation of TCs be distorted (since there is no data assimilation) such that different trackers would not necessarily find the stronger TCs, or not?

We removed the mention "in a simulation" at the end of the first sentence. Strictly speaking, we don't know whether this property hold in model simulations. As a starting point, we note that some of the properties found here for ERA5, for example that (i) UZ being more selective than TRACK and (ii) TRACK detecting genesis earlier than UZ hold well in models, are also found in models (Roberts et al. 2020a, 2020b). This gives some confidence that reanalysed and modelled cyclones share some properties. More analysis of TCs tracks obtained in simulations with trackers having different sensitivities could be performed to check whether they agree for the most intense TCs. That being said, we would rather assess these properties in a separate paper devoted to model simulations.

> **Comment 15**
>
> L581: "Contrasted results ...varying resolution of observations" – it is not clear to me what you are trying to say in these several sentences. Perhaps you could rewrite to make clearer? I really don't know what varying resolution of observations means, do you mean how frequent they have been over time, or how densely observed pre-/post- satellite era?

Mainly, we refer to the inhomogeneities in IBTrACS and ERA5. With varying resolution of observation we meant the density of observational points available for re-analysis. We suggest reformulating the sentence to remove the confusing "resolution":

> They may **result from inhomogeneities in IBTrACS** due to differences in reporting methods by each agency and/or **from inhomogeneities in ERA5** because of the varying **amount and density of** the observations available for assimilation.

> **Comment 16**
>
> L585: binarity – my dictionary does not have this word. Perhaps binary choice?

Indeed it was a poor translation on our side. Following you suggestion, we would change the sentence to

> Finally, it is important to keep in mind that deriving detection scores implicitly implies that there exists some sort of **binary choice between what is a TC or not.**

> **Comment 17**
>
> L587: "To some extent, trackers' thresholds are arbitrary" – this may be true, but could one also argue that observational classification of TCs is also somewhat arbitrary, satellite images with an interpreted core of the storm or other methods? Perhaps you disagree, but maybe you could qualify the statement a little.

I agree! I would say all that goes into our argument that these meteorological phenomena form a continuum, so that any attempt at categorizing them is intrinsically arbitrary.
We updated the text to clarify our point of view:

> Of course, this is not the case in reality where these meteorological systems form a continuum. **Hence, any attempt at categorizing them is intrinsically somewhat arbitrary. In the same way that classification based on satellite imagery or fixed wind thresholds is, to some extent, subjective,** trackers' thresholds are arbitrary and artificially create a strict limit in the gray zone that separates tropical disturbances, storms and cyclones.

> **Comment 18**
>
> L887: You could mention here that understanding precursors is becoming a key question for future projections (in particular to understand diverging projections of increase or decrease in frequency in future), so having trackers able to identify such features is really important.

We suggest the following addition to our text:

> [...] This property of TRACK **and OWZ** opens the way for studying the correlation between North Atlantic TC genesis and African easterly waves in ERA5, in the spirit of studies such as done by Thorncroft Hodges (2001), Hopsch et al. (2007) and Duvel (2021). It would be interesting to further

exploit that property by performing similar studies of TC precursors in other oceanic basins systematically, **especially in the context where some of the uncertainty related to climate change projections of TC activity is due to the lack of understanding of TC seeding and whether it is a driver of TCs natural variability (Vecchi et al. 2019).**
* * *
**Comment 19**

Table B1: For TRACK, I think here you are using the version averaging vorticity across 850,700,600 hPa, hence rather than eta_850 I think it should be eta_bar_T63, as you have written on L192.
* * *
Yes indeed. The text has been corrected accordingly.
* * *
**Comment 20**

Figure B2: What is the difference between Fig. B2 and the earlier, similar one, can you clarify?
* * *
This figure is the same that Figure 1, except it displays results for the VTU post-treatment. But this precision was indeed lacking in the description. We corrected this.
* * *
**Comment 21**

Looking at the algorithm parameters within StitchNodes, the different trackers have different minimum durations set. But duration (lifetime) of the tracks are compared in the results section. Would it not have made more sense to use the same minimum duration across all trackers?
* * *
Our choice for this paper was to use the trackers as they were already published and used. In this context, we agree that different minimum duration thresholds are one of the origins of the differences in outcome we see, especially considering the fact that we conclude that the main difference is different selectivity of weak and short track.

**2   Anonymous reviewer**
* * *
**Comment 1**

Abstract is missing the research problem of the article. Please include the goal of your research work in the abstract.
* * *
We added it at the end of the first part of the abstract :

The assessment of Tropical Cyclones (TC) statistics requires the direct, objective and automatic detection and tracking of TCs in reanalyses and model simulations. Research groups have independently developed numerous algorithms during recent decades in order to answer that need. Today, there is a large number of algorithms, often referred to as trackers, that aim to detect the positions of tropical cyclones in gridded datasets. **The questions we ask here are the following: does the choice of tracker impacts the climatology obtained? And, if it does, how should we deal with this issue?**

> **Comment 2**
>
> Paragraph 15 is not clear. Needs to explain the conclusion here more clearly. Does the author mean that we need to select one or a combination of few trackers with better performance and average the result.?

We suggest rephrasing the end of the abstract to :

> We conclude by advising against using as many trackers as possible . **We favor** a more efficient approach **involving the selection of** one or a few trackers with well-known properties.

> **Comment 3**
>
> Also, in the paragraph 75 the word "we used" has been used frequently, please reduce the usage of that. Also, in the entire draft we find the word "we used" is repeated. Instead use "we employed", "we utilized".

We reformulated the paragraph this way :

> The paper is organized as follows. After a description of the classification and datasets , we detail the algorithms of the four trackers  as well as our track matching method. We then use the four trackers to track TCs in ERA5 and to match the detected tracks with IBTrACS tracks, and we present a detailed analysis of the population of missing and false alarm tracks so obtained. This knowledge is **taken into account** to develop two methods common to all trackers that aim at filtering extra-tropical false alarms from the results. The filtered datasets are then used to analyze the sensitivity of traditional metrics to the choice of the trackers. Finally, we **gather** the insight gained from this analysis to consider the complementarity of different trackers and provide some guidelines for applying TC trackers to model results. The conclusion gives a summary of the trackers' common points and differences.

We also replaced several occurrences in the remainder of the manuscript.

> **Comment 4**
>
> At paragraph 40, you have mentioned that physics-based trackers embed a wind threshold. Please be precise here which wind threshold do you mean? I guess it should be 10m winds.

This modification has been applied:

> By contrast, the physics-based trackers usually embed a  threshold **on the 10m wind**, a parameter known to be very sensitive to resolution.

> **Comment 5**
>
> At the paragraph 45, mention that OWZ **tracker** was found to produce better results across a wide range of resolutions instead of just OWZ.

It has been corrected.

> **Comment 6**
>
> At line 70, Dulac et al., ? year is not mentioned here.

We wanted to refer here to a paper we thought would be published by the time of the acceptation of this one. However, due to delays, the paper is not yet submitted. We removed that reference and the associated sentence.
* * *
**Comment 7**

In the paragraph 130, the following sentence is not clear: Nevertheless, it has recently been assessed as having similar performances for a range of metrics (Zarzycki et al., 2021; Roberts et al., 2020a).
* * *
We added precision on the reanalyses at stake this way :

> Nevertheless, **ERA5** has recently been assessed as having similar performances **as JRA-55 or NCEP-CFSR** for a range of metrics (Zarzycki et al., 2021; Roberts et al., 2020a).
* * *
**Comment 8**

Coming to the methods section on TC trackers: It would be good to provide a table describing different tracker input variables, main idea of the design of the tracking scheme, spatial and temporal resolution requirements of the tracker variables, etc.
* * *
The different tracker input variable are detailed in Table B1. We are wary of synthetizing the idea of the design of a tracking scheme in a table, because we are unsure whether we can get the full grasp of its developer's mindset in one case. However, we can add a column with references to the relevant papers where the reader could find such information with more details. As for the temporal resolution requirement, there is no such explicit requirement for any of the trackers we used. Although we used 6-hourly data as was done in our reference versions of the trackers, some of them have been used and adapted for different temporal resolution, the other could also be adapted with little effort: given that the data is instantaneous wind speeds, there is no time-averaging that would require to modify the wind and/or vorticity thresholds; only adjusting the length on which the criteria are satisfied is necessary. As a matter of proof, following a question from the other referee, we tested tracking TCs with 1–hourly data and found very limited changes. For the spatial resolution, it is a trickier subject, and, as we already discussed in the introduction (end of the second paragraph), we know some tracker perform better than others at different resolutions. Unfortunately, a precise quantification of these ranges has yet to be performed. We added the following paragraph in the discussion section to highlight this problem :

> **Another consideration regards the resolution-(in)dependence of the tracking method, or its performance at a given target resolution. Here, the target resolution was that of ERA5, which is about $30\,\mathrm{km}$. The trackers we used either claim to be resolution-independent, or were calibrated on reanalyses with similar resolution, so that the target resolution here is supposedly optimal. It is not guaranteed that any of these trackers will behave similarly at resolution much lower or much higher that those of ERA5. In particular, trackers embedding a wind threshold might be particularly sensitive to resolution (walsh et al. 2013). There are also situations for which one would want to assess a set of simulations with a wide range of horizontal resolutions, and for which a resolution-independent method would be prefered. Even though there are arguments in the literature that dynamics-based trackers (e.g. TRACK, OWZ) might be less dependent on resolution than physics-based methodologies (Tory et al. 2013a, 2013b, Raavi et al. 2020), there is no quantitative assessment of this property. In general, we are lacking a quantification of the range of resolution for which trackers are valid, with or without retuning.**

> **Comment 9**
>
> In the description of the post treatment methods, where you have used two different methods and only focus mainly on the STJ method in the entire manuscript. So, I suggest maybe you can keep the detailed description explaining the VTU method in the supplementary section.

We prefer to keep both methods detailed in the body of the text, because we argue that this section is a contribution in itself and not just a method for the rest of the results. The two methods and complementary and have advantages and drawbacks that we want to illustrate, even though in our application they lead to the same results.

> **Comment 10**
>
> In the Discussion section, you have introduced Venn diagrams concept. I suggest you to give some details about how to build the Venn diagrams in the methods section.

Upon reading again the manuscript, we do realize that the way we had written it was interrupting the flow of the argumentation. Because they are not used for any scientific analysis per se, but rather for visualisation and in order to build a quick intuition on the different trackers properties, we don't think the description belongs to the methods. Instead, we moved them to the caption of the corresponding figure and we now specify the precise Python package we used to produce those Venn diagrams.